# Advice Querying under Budget Constraint for Online Algorithms

**Ziyad Benomar**
CREST, ENSAE, Ecole polytechique
ziyad.benomar@ensae.fr

**Vianney Perchet**
CREST, ENSAE and Criteo AI LAB
vianney.perchet@normalesup.org

## Abstract

Several problems have been extensively studied in the *learning-augmented* setting, where the algorithm has access to some, possibly incorrect, predictions. However, it is assumed in most works that the predictions are provided to the algorithm as input, with no constraint on their size. In this paper, we consider algorithms with access to a limited number of predictions, that they can request at any time during their execution. We study three classical problems in competitive analysis, the ski rental problem, the secretary problem, and the non-clairvoyant job scheduling. We address the question of when to query predictions and how to use them.

## 1   Introduction

With the rise of data science and the huge emphasis on research in machine learning and artificial intelligence, powerful predictive tools have emerged. This gave birth to *learning-augmented algorithms*, which use these predictions to go beyond the standard long-standing worst-case limitations. The design of such algorithms requires establishing good tradeoffs between consistency and robustness, i.e. having improved performance when the predictions are accurate, and not behaving poorly compared to the case without predictions if they are erroneous. This was formalized by Lykouris and Vassilvitskii [41] for the caching problem, and Purohit et al. [47] for the ski-rental and scheduling problems. Since then, the learning-augmented setting had many applications in implementing data structures [36, 23, 39] and in the design of algorithms [15, 7, 6, 11, 30, 5, 1, 4, 10, 40].

Several questions and research directions have been explored, such as exhibiting optimal robustness-consistency tradeoffs [48], incorporating predictions from multiple experts [3, 19], or customizing learning approaches to make predictions for such algorithms [49, 21, 18, 34, 2]. In most papers, those predictions are assumed to be given as inputs, which is unfortunately not realistic, practical or applicable to real-life scenarios. Indeed, many problems such as scheduling, paging, or set cover, require some arbitrarily large number of predictions (roughly speaking one per request or job). We claim that, on the contrary, algorithms might have only access to a finite budget of predictions as each one of them is costly to compute. Therefore, the main question rather becomes deciding how and when to spend the assigned budget, by requesting new predictions. Such a setting was considered for example in [31] for the caching problem. Moreover, it also happens that the quality of the predictions improves with time, as more data are collected, more knowledge of the problem is gathered, or delaying the computation of prediction allows to allocate more computational power to them, hence enhancing their precision, for instance with more powerful forecasting models. The question of planning the prediction requests is thus quite crucial.

The problems we mentioned fall into the category of online algorithms, operating under uncertainty, and gaining more information about the input of the problem through time, eventually depending on their past decisions. The performance of such algorithms is measured by their competitive ratio [14], which is the worst-case ratio between their output and that of the optimal offline algorithm, having full knowledge of the inputs and parameters of the problem. To illustrate our claims, we shall focus

37th Conference on Neural Information Processing Systems (NeurIPS 2023).

on three well-established and fundamental problems in competitive analysis: ski rental, secretary, and job scheduling problems.

**Ski-rental**  In the ski-rental problem, the decision maker faces each day the choice of either renting a ski for a unit cost, or buying it at a cost of $b$ and skiing for free after that. The only decision to be made by the algorithm is therefore when to buy. The unknown parameter is the length $x$ of the ski season, and the objective is to minimize the total cost of renting and buying. The optimal offline algorithm has a cost $\min\{x, b\}$, the deterministic online *break-even* algorithm, which consists in renting for the first $b-1$ days and then buying, results in a competitive ratio of 2 [32], and the optimal randomized algorithm introduced by Karlin et al. [33] yields a competitive ratio of $\frac{e}{e-1}$.

**Secretary problem**  The second problem we study is the secretary problem. Consider having $N$ applicants observed sequentially in a uniformly random order. Immediately after a candidate is observed, the algorithm has to either select them and stop, or refuse them irrevocably. The objective is to maximize the probability of selecting the best applicant overall. The particularity of this problem is that a single incorrect decision at any point leads to failure and zero gain. Consequently, the cost of making a wrong decision is significantly high. The optimal offline algorithm has access to all values and thus can choose the maximum with probability 1. The optimal online strategy is to reject the first $N/\mathrm{e}$ applicants, then stop at the first applicant who is better than all of them [17, 25]. This is often referred to as the $1/\mathrm{e}$ rule, and it gives a success probability of $1/\mathrm{e} \approx 0.368$.

**Job scheduling**  The third problem we consider is the preemptive non-clairvoyant job scheduling. The input is a set of jobs with unknown sizes, that the algorithm needs to schedule on a single machine, with the possibility of halting a job during its execution and resuming it later. The goal is to minimize the sum of their completion times, i.e. finish as many jobs as possible rapidly. The only observations made during the run of an algorithm are the completion times of the jobs. These observations, along with the times at which they are received, depend on both the input instance and the algorithm's past decisions. The cost of early incorrect decisions is more significant than that of later incorrect decisions. Take for example the instance of job sizes $\{x_1 = 1, x_2 = 100\}$, running $x_1$ then $x_2$ gives an output $1 + 101 = 102$, while running them in the inverse order gives $100 + 101 = 201$. The optimal offline algorithm is to schedule the jobs in increasing order of size, and a deterministic online algorithm called *round-robin*, gives a competitive ratio of 2, which is optimal for an arbitrarily large number of jobs [46].

## 1.1  Organization and contributions

In Section 2, we consider the ski rental problem, where we assume that an oracle provides binary predictions on how the snow season length $x$ compares to the purchasing cost $b$, and that these predictions are accurate with a probability that is a function of time, known to the algorithm and denoted as $p_t$. This assumption accurately reflects and models what is practically occurring. As the snow season begins, there is significant uncertainty about how long it will last. However, over time, the accuracy of predictions increases as more weather observations are made, the direction of snowstorms becomes clearer, and additional data is collected. The algorithm is permitted to request only one prediction during its execution, at a time of its choice. Depending on the function $p_t$ and on how the prediction is used once queried, it can be more advantageous to rent for a period of time $t$ before requesting it, i.e. paying a cost for making a better decision in the future. We consider that once the prediction is queried, we use either the deterministic or the randomized algorithm presented in [47] for the ski rental problem. We show how to optimally choose $t$, and how the knowledge of $p_t$ allows to tune the parameter $\lambda$, regulating the levels of consistency and robustness in both algorithms.

Secondly, we consider in Section 3 the secretary problem with access to an oracle, accurate with a probability $p$, telling if there are better applicants in the future. In practice, this would correspond to the following scenario: consider a decision-maker with an imperfect interviewing process, and thus can only compare the applicants with each other, i.e. observe their relative ranks, but lacks access to their true value. On the other hand, an expert who, due to extensive experience, knows how to better interview the applicants and extract their true value, and also knows the distribution of their values (as in a prophet setting with i.i.d variables). The decision-maker can ask the expert to interview some applicants for a cost. After that, the expert does not disclose the precise value of the applicant - that is irrelevant to the decision maker-, but they provide a recommendation (accept or reject) and

accompany it with a confidence level, which is the probability that this recommendation is accurate. Assuming that the decision-maker has enough budget to ask for the expert's advice $B$ times, we give an algorithm whose success probability depends on $p$ and $B$, converges to 1 as $p$ goes to 1 and $B$ grows, and that is always lower bounded by $1/e$. Our algorithm easily adapts to the case where the accuracy of the oracle is time-dependent.

Section 4 is dedicated to the preemptive non-clairvoyant job scheduling, where we assume that the algorithm can request access to the sizes of $B$ chosen jobs. In contrast with the two previous problems, there is no advantage in waiting before querying the hints, and since the jobs play identical roles, the algorithm can only request the sizes of $B$ randomly chosen ones. We show first that the competitive ratio 2 cannot be improved unless $B = \Omega(N)$, where $N$ is the number of jobs. Then we present an algorithm that chooses randomly $B$ jobs and requests their sizes, then runs concurrently the optimal offline algorithms on those jobs and round-robin on the others, with time-dependent processing rates. We prove a generic expression for the output of this algorithm that depends on the chosen processing rate, and then we give a particular processing rate that yields a competitive ratio of $2 - (B/N)^2$, therefore interpolating the competitive ratios 2 and 1 met respectively when no hint is given, and when all the job sizes are known.

Finally, we run simulations of our algorithms in Section 5. We show that the performance of the algorithm we introduced for the secretary problem matches our theoretical lower bound, and we compare it to another heuristic version that we did not study theoretically. Then, by exhaustively testing the algorithm we designed for the scheduling problem with access $B$ job sizes on various benchmark instances, we observe that it has a better performance in practice than the theoretical upper bound $2 - (B/N)^2$.

## 1.2 Related work

The ski-rental and scheduling problems have received extensive attention in the realm of learning-augmented algorithms [47, 48, 3, 19, 18, 37, 9]. They serve respectively as notable examples for problems requiring a single prediction and multiple predictions. Moreover, they both have numerous applications and variants, as [32, 12, 24, 44] for the ski rental and [42, 16, 38] for the scheduling problem. The secretary problem was also among the first problems studied with the advice model. Dütting et al. [22] consider that a binary prediction is received with each applicant, indicating whether it is the best overall or not. Assuming that these predictions are accurate each with a probability $p$ independently, they design algorithms that improve upon the $1/e$ success probability. While this model closely aligns with ours, we consider a scenario where the algorithm has a limited budget of predictions and must carefully determine when to query them. Antoniadis et al. [7] studied a variant of the problem where the objective is to maximize the value of the selected applicant. They consider that the algorithm is provided with a prediction of the value of the best applicant.

For both the ski-rental and the secretary problems, we consider predictions that are accurate with a known probability. A similar assumption was made in [28], where the oracle is assumed to deliver a prediction that is accurate with a probability of at least $\epsilon$, and that can be arbitrarily inaccurate with the remaining probability. The authors give improved competitive ratios depending on $\epsilon$ for some online problems including caching, online set cover, and facility location.

The aspect of having a limited prediction budget is relatively underexplored in the literature. The question was initially examined in the context of the online linear optimization problem with hints, where Bhaskara et al. [13] demonstrated that a sublinear number of hints is sufficient to achieve regret bounds similar to those in the full hints setting when the timing of requesting hints is well chosen. Im et al. [31] also investigated this question for the caching problem, presenting an algorithm that strategically utilizes the assigned prediction budget to improve the competitive ratio as the budget increases. In a very recent paper, the ski-rental and the Bahncard problems were explored in a setting with costly predictions [20], where the cost of predictions is added to the total cost paid by the algorithm. This can be seen as a penalized version of querying predictions under budget constraints. Other works have also investigated the reduction of the prediction size rather than the number of predictions. Specifically, they consider binary predictions encoded on a single bit and explore how the competitive ratio can be enhanced compared to other types of predictions [22, 45, 8].

Another important question we cover thanks to the ski-rental problem is how to optimally balance robustness vs consistency with respect to predictions [47]. This tradeoff is usually done by adding

some extra-parameter $\lambda \in [0, 1]$ that reflects how much the decision-maker is willing to trust blindly the prediction, and which appears naturally in the bounds of the competitive ratio. Roughly speaking, for $\lambda = 1$, the decision maker focuses solely on the performances of their algorithms when predictions are incorrect (hence predictions are actually disregarded), while for $\lambda = 0$, they naïvely consider that predictions are correct. Intermediate values of $\lambda$ correspond to less extreme behaviors, and the final competitive ratio strongly depends on the value of this parameter and the total "amount" of errors (measured with respect to some problem-dependent metric) in predictions. Setting a value for this parameter requires having some knowledge about the quality of the prediction. Khodak et al. [34] shows how to learn to set $\lambda$'s value in an online learning setting, where the algorithm runs repeatedly on different instances, and learns to predict unknown parameters based on features on the new instances. After many runs, the value of $\lambda$ can be increased since the predictions become more and more accurate. In our case, we have binary predictions that are correct with a known probability, which is a natural assumption for binary predictions [45, 22], and we show that this allows us to optimally choose the value of $\lambda$.

## 2  Ski-rental with time-dependent guarantees on the prediction

We consider that the cost of renting a ski for one day is 1, while the cost of buying is $b > 1$, and we denote $x$ the duration of the ski season, which is unknown. For all the algorithms we present, if the ski season is over then the algorithm stops and no further cost is paid. In the learning-augmented setting, we assume that the algorithm possesses a prediction $y$ for $x$. Many variants of this scenario have also been explored in previous research papers [47, 35, 27, 48]. We restrain ourselves to the case of binary predictions, comparing the number of snow days to the budget. More precisely, we assume the existence of an oracle, that can be called at any time $t$, predicting whether $x - t > b$ or not, where $x - t$ is the number of remaining snow days. Furthermore, we assume that the accuracy of the oracle improves over time. If queried at time $t$, then the prediction is correct with a probability $p_t$, known to the algorithm, that is independent of the problem's history and increases over time. We assume that, due to budget limitations, the algorithm can access the oracle only once during its execution, and thus it must carefully choose the time of asking for the prediction.

Let $\mathsf{ALG}$ be an algorithm such that, with a prediction accurate with probability $p$, it has a competitive ratio $C(\mathsf{ALG}, p)$. We define the algorithm $\mathsf{ALG}_t$ that rents for the first $t$ days, then queries a prediction $\mathcal{Q}_t$ of $\mathbb{1}(x - t \geq b)$ at the start of day $t + 1$, and then acts like $\mathsf{ALG}$. We have the following result.

**Lemma 2.1.** *$\mathsf{ALG}_t$ has a competitive ratio of at most $\frac{t}{b} + C(\mathsf{ALG}, p_t)$.*

The term $t/b$ represents the additional cost due to renting the first $t$ days, in order to have a better accuracy $p_t$ which decreases the second term. The optimal time for querying a prediction is $t^\star$ minimizing the function $t/b + C(\mathsf{ALG}, p_t)$. Although this requires knowing all the sequence $(p_t)_{t \geq 0}$ in advance, we can design simple online heuristics for being close to a local minimum, where the accuracy of the oracle at some step $t$ is only revealed when that step is reached. We can for example access the oracle at the first time $t$ when $t/b + C(\mathsf{ALG}, p_t)$ increases.

In the following, we show how Lemma 2.1 can be applied with explicit algorithms $\mathsf{ALG}$ that are given a binary prediction as input. We consider the algorithms 1 and 2, which were first introduced in [47]. In both algorithms, the parameter $\lambda$ indicates how much the prediction is trusted. Assuming that the input prediction is accurate with a probability $p = \mathbf{P}(\mathcal{Q} = \mathbb{1}(x \geq b))$, we show how to tune $\lambda$ in both algorithms to minimize their costs, and we upper bound their competitive ratios.

---

**Algorithm 1:** Deterministic algorithm with input binary prediction [47]

**Input:** cost $b$ for buying, a prediction $\mathcal{Q}$ for $\mathbb{1}(x \geq b)$

1 **if** $\mathcal{Q} = 1$ **then** buy on the start of day $\lceil \lambda b \rceil$ ;
2 **else** buy on the start of day $\lceil b/\lambda \rceil$ ;

---

**Lemma 2.2.** *If the oracle $\mathcal{Q}$ delivers an accurate prediction with probability $p \geq 0.5$, then by choosing $\lambda = \sqrt{\frac{1-p}{p}}$, the algorithm achieves a competitive ratio of at most $1 + 2\sqrt{p(1-p)}$.*

The optimal choice of $\lambda$ gives therefore a competitive ratio that is always upper bounded by 2, which is the optimal competitive ratio without prediction, and decreases to 1 when the accuracy of the oracle

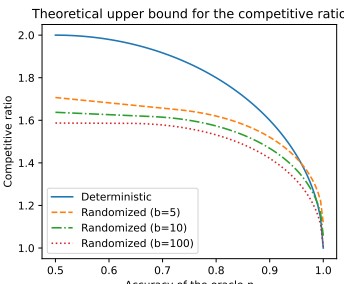

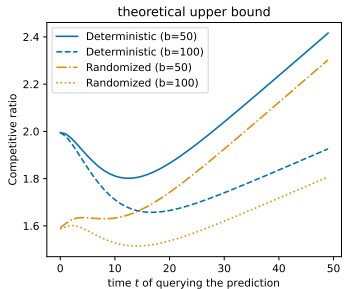

Figure 1: Competitive ratios of Algorithms 1 and 2 with the optimal choice of $\lambda$, for $p \in (0.5, 1)$ and $b = 5, 10, 100$

Figure 2: Competive ratio obtained by renting $t$ days before querying a prediction, with $p_t = 0.95 - 0.4 \exp(-t/5)$ and $b = 50, 100$

is better. While Algorithm 1 is deterministic, the next algorithm we study is randomized, where the day of buying is a random variable, drawn from a probability distribution that depends on the prediction $\mathcal{Q}$. We show again an optimal choice of $\lambda$ for minimizing the competitive ratio.

---

**Algorithm 2:** Randomized algorithm with input binary prediction [47]

**Input:** cost $b$ for buying, a prediction $\mathcal{Q}$ for $\mathbb{1}(x \geq b)$

1 **if** $\mathcal{Q} = 1$ **then** buy on the start of day $d \in \{1, \ldots, \lfloor \lambda b \rfloor\}$, with probability $\propto (1 - 1/b)^{\lfloor \lambda b \rfloor - d}$ ;
2 **else** buy on the start of day $d \in \{1, \ldots, \lceil b/\lambda \rceil\}$, with probability $\propto (1 - 1/b)^{\lceil b/\lambda \rceil - d}$ ;

---

**Lemma 2.3.** *If the oracle $\mathcal{Q}$ delivers an accurate prediction with probability $p \geq 0.5$, then with $\lambda = \min\left\{1, 1/b + \sqrt{(e-1)(1/p - 1)}\right\}$, Algorithm 2 has a competitive ratio of at most*

$$\begin{cases} \frac{1}{e-1} + 2\sqrt{\frac{p(1-p)}{e-1}} + \left(1 - \frac{1-1/b}{e-1}\right)p & \text{if } p \geq \left(1 + \frac{(1-1/b)^2}{e-1}\right)^{-1}, \\ \frac{e}{e-1} + \frac{1-p}{b-1} & \text{otherwise.} \end{cases}$$

Figure 1 shows the competitive ratios of both algorithms depending on $p$ when $\lambda$ is chosen optimally. The upper bound shown in Lemma 2.3 depends on $b$, this is why we test it with different values of $b$. As expected, the randomized algorithm yields better guarantees than the deterministic one. However, when $p$ is very close to 1, Algorithm 1 is slightly better, because the randomized algorithm requires having $\lambda \geq 1/b$, and therefore in the limit where $p = 1$, the optimal choice is $\lambda = 1/b$, giving the upper bound $1 + 1/b$ instead of 1 on the competitive ratio.

Now, assuming that the oracle is accurate with a known time-dependent probability $p_t$, the previous lemmas allow to optimally choose the time of querying the prediction when one of the two algorithms we presented is used after the prediction is obtained. We only state the result for the case where Algorithm 1 is used. The proof is immediate using Lemmas 2.1 and 2.2, and a similar result can be shown using Lemmas 2.1 and 2.3 when Algorithm 2 is used instead.

**Theorem 2.4.** *If the predictions delivered by the oracle are accurate with a probability $p_t$ that only depends on the time, then renting until time $t$, then querying a prediction $\mathcal{Q}_t$ and running Algorithm 1 with parameter $\lambda = \sqrt{\frac{1-p_t}{p_t}}$, yields a competitive ratio of at most $1 + t/b + 2\sqrt{p_t(1-p_t)}$.*

We consider in Figure 2 an example where $p_t = 0.95 - 0.4 \exp(-t/5)$, thus $p_0 = 0.55$ and $\lim_{t \to \infty} p_t = 0.95$. The figure shows the competitive ratio of renting $t$ days then using one of the Algorithms 1 or 2, with $b = 50, 100$. We observe that this strategy can significantly improve the competitive ratio if $t$ is chosen correctly. Of course, this depends strongly on $p_t$ and $b$. In particular, for the randomized algorithm, it is better to query the prediction at $t = 0$ when $b = 50$.

## 3 Secretary problem with $B$ predictions

Assume that $N$ applicants are observed sequentially in a uniformly random order $(x_1, \ldots, x_N)$, all having distinct values. After an applicant is interviewed, the decision-maker has to either accept them

and halt the process, or refuse them irrevocably. We consider a setting where, when an applicant $x_t$ is observed, and the budget is still not exhausted, the algorithm can request a binary prediction $\mathcal{Q}(x_t)$ indicating if there are better applicants coming in the future. We assume a maximal budget of $B$ predictions, where $B$ is a constant independent of $N$. We assume first that the predictions are error-free, and we give theoretical guarantees on Algorithm 3 in that case, and after that, we show how it can be adapted to handle predictions that are accurate only with a probability $p$. In this section, we say that an algorithm succeeded if the selected applicant is the best overall.

We consider first Algorithm 3, which rejects the first $\lceil r_B N \rceil$ applicants, where $r_B \in (0, 1)$ is a threshold depending on $B$, and then queries a prediction for the first applicant $x_t$ better than all of them. If the oracle predicts that there is a better candidate in the future ($\mathcal{Q}(x_t) = 1$) then the algorithm is restarted with the inputs $(x_{t+1}, \ldots, x_N)$ and budget $B - 1$.

---

**Algorithm 3:** ADATHRESH Adaptive Threshold

---

**Input:** Budget $B$, sequence of applicants $(x_1, \ldots, x_N)$
1 Reject the first $\lceil r_B N \rceil$ applicants;
2 **for** $t = \lceil r_B N \rceil + 1, \ldots, N$ **do**
3     **if** $x_t > \max\{x_1, \ldots, x_{t-1}\}$ **then**
4         **if** $B > 0$ **then**
5             query a prediction $\mathcal{Q}(x_t)$;
6             **if** $\mathcal{Q}(x_t) = 0$ **then** Return: $t$ ;
7             **else** Return ADATHRESH$(B - 1, (x_{t+1}, \ldots, x_N))$;
8         **else** Return: $t$;

---

**Theorem 3.1.** *Let* $(q_B)_{B \geq -1}, (r_B)_{B \geq 0}$ *the sequences defined by* $q_{-1} = 0$ *and for* $B \geq 0$

$$q_B = q_{B-1} + (1 - q_{B-1}) \exp\left(-\frac{1}{1 - q_{B-1}}\right), \quad r_B = \exp\left(-\frac{1}{1 - q_{B-1}}\right).$$

*If the oracle* $\mathcal{Q}$ *delivers error-free predictions, then with the thresholds* $(r_B)_B$, ADATHRESH *with budget* $B$ *has a success probability of at least* $q_B$ *independently of the input size* $N$.

Observe that, as $B$ increases, the lower bound $q_B$ of the success probability of ADATHRESH converges to 1 and the thresholds $r_B$ converge to zero, meaning that the higher the budget, the higher the risks: there are fewer applicants in the first observation phase, thus the probability of selecting a future sub-optimal one is higher (yet this risk is hedged by the predictions), but on the other hand, it reduces the probability of naïvely disregarding the best applicant if it is among the first arriving ones. Although ADATHRESH is a naive algorithm that does not make full use of past information, we showed that adequately choosing the thresholds $(r_B)_B$ guarantees a success probability that goes to 1 as the budget increases. We present in Appendix B.3 an improved version of ADATHRESH that keeps in memory the maximum value $M$ observed so far and that rejects all applicants having values below $M$. We show numerically in Section 5 how this increases the success probability of the algorithm.

**Handling imperfect predictions** If the predictions of the oracle are imperfect, then one way of guaranteeing robustness, since the objective is to maximize the success probability, is to use the $1/e$-rule with probability $\lambda$ and Algorithm 3 with probability $1 - \lambda$. This guarantees a success probability of at least $\lambda/e$ if the predictions are incorrect. On the other hand, given that the oracle's predictions are binary, it is reasonable to assume that each prediction is correct with a probability $p$ independent of all the observations, as explained in the introduction. Based on this assumption, Algorithm 3 can be modified to include the option of trusting or disregarding the oracle's prediction at each step. We show that it is advantageous to trust the oracle when $p$ is above a certain threshold and run the $1/e$-rule otherwise. This algorithm achieves a success probability of at least $1/e$, with improving performance as $p$ approaches 1. We show in the appendix how this can be generalized when the accuracy of the oracle is time-dependent.

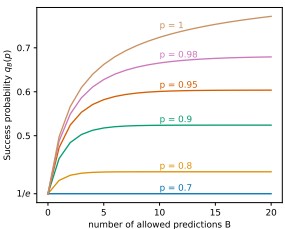

Figure 3: $(q_B(p))_{B,p}$ for $p \in [0.5, 1]$ and $B \leq 20$

**Theorem 3.2.** *Assume that each oracle's prediction is independently correct with probability* $p \geq \left(1 + (e-1)\exp(-\frac{e}{e-1})\right)^{-1} \approx 0.73$, *then there is an algorithm that achieves a success probability of at least* $q_0(p) = 1/e$ *for* $B = 0$, *and* $q_B(p)$ *for* $B \geq 1$ *defined as*

$$q_B(p) = pq_{B-1}(p) + p(1 - q_{B-1}(p))\exp\left(-\frac{1}{1-q_{B-1}(p)}\right).$$

*If we denote* $W$ *the Lambert function, i.e. the inverse of* $u \mapsto ue^u$ *on* $(0, \infty)$, *then we have*

$$\lim_{B \to \infty} q_B(p) = 1 - \frac{1}{1 + W\left(\frac{p}{e(1-p)}\right)}.$$

*Moreover, the algorithm has a success probability of at least* $1/e$ *for any value of* $p \in [1/2, 1]$.

**Optimal algorithm** Assuming that the predictions are error-free, the problem is equivalent to the *multiple-choice secretary problem* [26], where the algorithm is allowed to select $k \geq 1$ applicants, and it is successful if the best overall candidate is among them. Indeed, the algorithm is given $B + 1$ attempts to identify the best applicant, and it halts if it finds it. This is analogous to choosing $k = B + 1$ applicants, where following the selection of each one, the algorithm employs a selection strategy as if the previous guesses were unsuccessful. The optimal algorithm for selecting $k$ applicants is a $(a_k, \ldots, a_1)$-rule [26, 43], where at any step $t$, if the number of applicants already selected is $i \in \{0, \ldots, k\}$, then reject everyone until step $\min\{t, a_{k-i}\}$, and accept the first applicant after that is the best observed so far. Although this family of algorithms has a simple structure, analyzing it is difficult and hides many technical challenges. The optimal thresholds and the asymptotic success probability are explicitly computed only up to $k \leq 5$, and a recursive formula is proven, via a dynamic programming approach, to compute the next thresholds. This formula is however difficult to exploit even numerically, while the optimal thresholds and the success probability of ADATHRESH can be computed very easily.

If the predictions are correct with a probability $p$, then the optimal algorithm must be a generalization of the $(a_k, \ldots, a_1)$-rule, and thus it is even harder to analyze. The algorithm we proposed is naive in the sense that it does not take into account all the past information and only remembers the history since the last prediction was queried. However, it illustrates how the limited budget of predictions should be spent, and it presents good robustness and consistency guarantees with respect to $p$ and $B$, in the sense that it always has a success probability of at least $1/e$, which is optimal without predictions, and it has a success probability that converges to 1 as $B$ and $p$ increase.

## 4 Preemptive $B$-clairvoyant job scheduling

We consider the problem of scheduling multiple jobs on a single machine, with the objective of minimizing the sum of their completion times. More particularly, we place ourselves in the *preemptive* setting, where the jobs can be temporarily halted and resumed later, or equivalently that they can be run in parallel with rates that sum at most to 1. Let $N$ be the number of jobs and $x_1, \ldots x_N$ their sizes. If the algorithm knows beforehand the sizes of the jobs, it is called *clairvoyant*, and the optimal algorithm OPT is to run the shortest jobs first. An algorithm is *non-clairvoyant* if the size $x_j$ of any job $j$ is unknown until the job is completed. Motwani et al. [46] showed that no deterministic or randomized algorithm can have a better competitive ratio than 2, which is achieved by *round-robin* (RR), which is a deterministic algorithm. RR works as follows: at any time $t$, if $n$ is the number of remaining jobs, then RR runs them all in parallel with rates $1/n$ each.

An in-between setting, surprisingly not explored yet, is when the algorithm has only access to the sizes of a limited number of jobs. We say that an algorithm is $B$-clairvoyant when it is allowed to access the sizes of $B$ jobs. We assume that it can query their true sizes, and not just noisy predictions. We start with a negative result, stating that we need to have $B = \Omega(N)$ to achieve a better competitive ratio than 2. Then, we give an algorithm with a competitive ratio of at most $2 - (B/N)^2$.

Let us first remind a few classic notations in the scheduling problem. For any algorithm ALG, and any instance $\mathcal{J} = \{x_1, \ldots, x_N\}$, the sum of the completion times obtained by ALG can be written as

$$\mathsf{ALG}(\mathcal{J}) = \sum_{i=1}^{N} x_i + \sum_{i<j} \left(D^{\mathsf{ALG}}(i,j) + D^{\mathsf{ALG}}(j,i)\right), \tag{1}$$

where $D^{\mathsf{ALG}}(i,j)$ is the delay caused by job $i$ to job $j$, i.e. the amount of job $i$ executed before the completion of job $j$. With this notation, we have for any $i < j$ that $D^{\mathsf{OPT}}(i,j) + D^{\mathsf{OPT}}(j,i) = \min\{x_i, x_j\}$ and $D^{\mathsf{RR}}(i,j) = D^{\mathsf{RR}}(j,i) = \min\{x_i, x_j\}$. Thus, if $x_1 \leq \ldots \leq x_N$ we obtain

$$\mathsf{OPT}(\mathcal{J}) = \sum_{i=1}^{N} x_i + \sum_{i=1}^{N} (N-i)x_i \quad , \quad \mathsf{RR}(\mathcal{J}) = \sum_{i=1}^{N} x_i + 2\sum_{i=1}^{N} (N-i)x_i. \tag{2}$$

## 4.1 Few hints are not enough

In opposite to other problems where it has been proved that a sublinear number of hints is enough for improving the performance [13, 31], the following theorem demonstrates that, for the scheduling problem, no algorithm can achieve a competitive ratio better than 2 when $B = o(N)$.

**Theorem 4.1.** *Any $B$-clairvoyant deterministic or random algorithm with $B = o(N)$, has a competitive ratio lower bounded by 2.*

## 4.2 Parallel OPT-RR algorithm with adaptive processing rates

We assume in this section that the $B = \Omega(N)$. A first naive algorithm would run $\mathsf{RR}$ until there are $B$ jobs left, then query their sizes, and use $\mathsf{OPT}$ to finish. However, when all the jobs have the same size, they terminate at the same time and no hint is queried. The output of the algorithm with this instance is exactly the same as $\mathsf{RR}$, which is twice the output of $\mathsf{OPT}$ asymptotically in $N$ [46], and therefore its competitive ratio is at least 2. More generally, any algorithm that runs $\mathsf{RR}$ waiting for a certain number of jobs to finish before querying the sizes of $B$ unfinished ones is no better than $\mathsf{RR}$ for the same reasons. Alternatively, the algorithm can wait for a possibly random amount of time $T > 0$, independent of the observed completion times, before querying the first hint. However, by taking job sizes sampled from an exponential distribution with parameter $\mu$ such that their sum is smaller than $T$ with an arbitrarily high probability, the algorithm terminates with high probability before requesting any hint and is no better than a non-clairvoyant algorithm on such input instances, leading to a competitive ratio of at least 2. Therefore, the best moment to query the hints is at the very beginning of the execution, and since the algorithm cannot differentiate between the jobs, it can only query the sizes of $B$ randomly chosen ones.

We propose a generic algorithm that queries the sizes of $B$ randomly chosen jobs, then concurrently runs $\mathsf{OPT}$ on them and $\mathsf{RR}$ on the others with respective rates $\alpha$ and $1 - \alpha$, where $\alpha$ is a parameter that can be adjusted throughout the course of the algorithm, depending on the information available at each time, i.e the predicted job sizes, the number of the remaining jobs, and the sizes of finished jobs.

---

**Algorithm 4:** $\mathsf{PAR}$ Parallel algorithm with Adaptive processing Rate

**Input:** Budget $B$, $N$ jobs with unknown sizes $\{x_1, \ldots, x_N\}$
1  $I \leftarrow$ Sample $B$ jobs uniformly at random without replacement;
2  $J \leftarrow \{1, \ldots, N\} \setminus I$ ;
3  **while** *there are still unfinished jobs* **do**
4  $\quad$ Adjust $\alpha$;
5  $\quad$ **run for a time unit**
6  $\quad\quad$ with rate $\alpha$: $\mathsf{OPT}$ on $\{x_i\}_{i \in I}$ ;
7  $\quad\quad$ with rate $1 - \alpha$: $\mathsf{RR}$ on $\{x_j\}_{j \in J}$ ;

---

In the following, for any algorithm $\mathsf{ALG}$ and for any subsets $H, K$ of $\{1, \ldots, N\}$, we denote $D^{\mathsf{ALG}}(H, K) = \sum_{i \in H} \sum_{j \in K \setminus \{i\}} D^{\mathsf{ALG}}(i,j)$ the sum of all the delays caused by jobs in $H$ to those in $K$. We demonstrate first a generic upper bound for the output of $\mathsf{PAR}$,

**Lemma 4.2.** *For any update rule of $\alpha$ we have*

$$\mathbf{E}[\mathit{PAR}(\mathcal{J})] = \sum_{i=1}^{N} x_i + \left(2 - \frac{B}{N}\left(4 - 3\frac{B-1}{N-1}\right)\right) \sum_{i=1}^{N} (N-i)x_i + \mathbf{E}[D^{\mathit{PAR}}(I, J) + D^{\mathit{PAR}}(J, I)].$$

This Lemma shows that the output of $\mathsf{PAR}$ depends only on the delays generated by jobs in $I$ on jobs in $J$ and vice versa. The difficulty now is to choose an adequate update rule for the rate $\alpha$, with provable upper bounds on $\mathbf{E}[D^{\mathsf{PAR}}(I, J) + D^{\mathsf{PAR}}(J, I)]$.

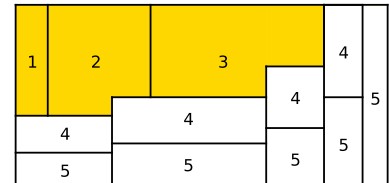

Figure 4: In this example, $N = 5$, the job sizes are $\{x_1 = 1, x_2 = 3, x_3 = 4, x_4 = 5, x_5 = 6.3\}$, and $I = \{1, 2, 3\}$. On the left we visualize a run of RR on this instance, and on the right a run of PAR with the SRR update rule. The X-axis represents time and the Y-axis represents the processing power allocated to each job.

### 4.3 Simulated round-robin update rule

We consider the *Simulated Round-Robin* (SRR) update rule, which adjusts $\alpha$ as follows. It simply puts a global processing rate on $I_t$, the set of unfinished jobs in $I$, proportional to the cardinal of $I_t^{\mathsf{RR}}$, the number of unfinished jobs in $I$ had RR be run instead (this counterfactual quantity can be computed at any time step as predictions are correct, see proof of Theorem 4.3) and similarly for $J$. Formally, $\alpha_t^{\mathsf{RR}} = |I_t^{\mathsf{RR}}|/(|I_t^{\mathsf{RR}}| + |J_t^{\mathsf{RR}}|)$. Figure 4 gives an illustration of the SRR update rule: to the left, it shows a run of RR, where initially the 5 jobs run each with a processing rate $1/5$ until job 1 terminates, then the remaining jobs run each with a processing rate of $1/4$ until job 2 terminates, and so on. To the right, the figure shows a run of PAR with SRR update rule, where the algorithm knows the sizes of jobs 1,2 and 3. The total processing rate of these jobs during the run of the algorithm is represented by the yellow area, and it is identical to their total processing rate during the run of RR.

**Theorem 4.3.** *If PAR uses a processing rate $\alpha_t^{\mathsf{RR}}$, then it is at most $\left(2 - \frac{B(B-1)}{N(N-1)}\right)$-competitive.*

The competitive ratio of PAR with processing rate $(\alpha_t^{\mathsf{RR}})_{t \geq 0}$ decreases as $B$ grows, going from 2 to 1, thus interpolating the non-clairvoyant and the clairvoyant settings.

## 5 Experiments

In this section, we test the performance of the algorithms we presented for the ski-rental, secretary, and scheduling problems, supporting our theoretical results and giving further insight.

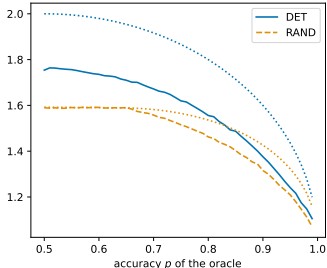

Figure 5: Competitive ratios of Algorithms 1 and 2, with an oracle $\mathcal{Q}$ accurate with probability $p \in [0.5, 1]$ and $b = 50$.

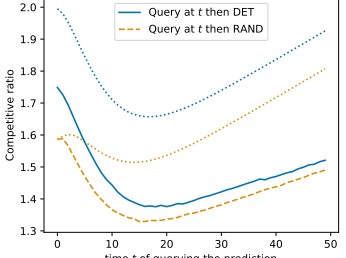

Figure 6: Competitive ratio of querying the prediction at time $t$, then running Algorithm 1 or 2, with $p_t = 0.95 - 0.4 \exp(-t/5)$ and $b = 100$.

For the ski-rental problem, we set a buying cost of $b = 50$ for Figure 5 and $b = 100$ for Figure 6, and the number of snow days is sampled randomly from a uniform distribution in $[1, 4b]$. Each point in both figures is computed over $10^5$ simulations. The value of $\lambda$ is chosen optimally with respect to $p$ as indicated in Lemmas 2.2 and 2.3.

The competitive ratios of both algorithms 1 and 2 when the prediction is given as input are shown in Figure 5, as well as their theoretical upper bounds. The experimental ratio of the deterministic

algorithm in this particular scenario is significantly better than the theoretical upper bound, while that of the randomized algorithm is close to the theoretical upper bound. In Figure 6, we consider that the oracle is correct with a time-dependent probability $p_t = 0.95 - 0.4 \exp(-t/5)$. We show the competitive ratios obtained by renting until time $t$ then querying a prediction and running the deterministic or the randomized algorithm. The theoretical upper bounds are represented in dotted lines. We observe that adequately choosing the time of querying the prediction can significantly improve the competitive ratio, which proves our claims. This time strongly depends on the cost $b$ and on the evolution of the probability $p_t$ with time.

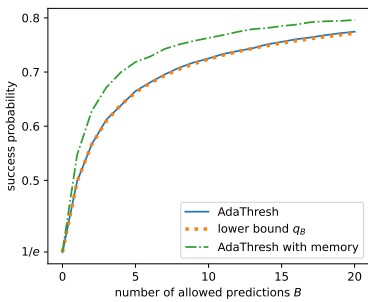

Figure 7: Success probability of ADATHRESH compared to the lower bound $q_B$, and compared to the case with memory.

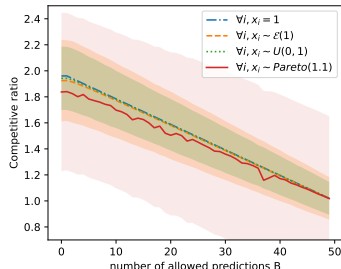

Figure 8: Competitive ratio of PAR with SRR update rule, tested with $N = 50$ and different benchmark input instances.

In Figure 7, we test ADATHRESH with $N = 1000$, and we show how the success probability improves with $B$. We also test a variant of the algorithm memorizing the value of the best-observed applicant after restarting, which improves the success probability (See Appendix B.3 for a detailed description of the algorithm). The success probability of ADATHRESH matches $q_B$, therefore it is a tight lower bound. We also observe that the success probability increases rapidly for the first values of $B$, but then becomes slower. This is because whenever the algorithm is restarted, there is a new observation phase, with a risk of missing the best applicant.

Secondly, we test the algorithm PAR with the SRR update rule on various benchmark inputs. We test it with $N = 50$ and jobs having (i) identical sizes, (ii) sizes sampled from the exponential distribution, (iii) uniform distribution, (iv) and Pareto distribution. (i) is a critical instance because it is the worst-case input for RR. (ii) is a classical benchmark used in many variants of the scheduling problem to prove lower bounds [46]. (iii) is a natural benchmark to test randomness. Finally, (iv) is well-suited in practice for modeling the job size distributions [47, 29], and also it shows how the algorithm behaves on instances with very high variance. Each point in the figure was obtained by averaging over 10000 runs. we see that the competitive ratio of PAR is at most $2 - B/N$ for all these benchmarks, which is better than the upper bound $2 - (B/N)^2$ proved in Theorem 4.3. Therefore, PAR can be highly efficient in many cases, since the gain obtained with $B$ known job sizes is proportional to $B$ as shown in the simulations.

## 6 Conclusion

We presented different settings where online algorithms operate under a restricted budget of predictions, that can be queried during their execution. We show that adequately using this budget significantly improves upon the worst-case performance. Our results pave the way for investigating more realistic and practical challenges within the learning-augmented paradigm, providing insights that can be extended to various other problems in competitive analysis.

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

# Advice Querying under Budget Constraint for Online Algorithms

## Omitted proofs

## A   Ski-rental with time-dependent guarantees on the prediction

### A.1   Proof of Lemma 2.1

*Proof.* In all the following, We denote $A(I)$ the output of algorithm A when given an input instance $I$. We will prove the claim of the lemma by treating separately the cases where $x \geq b$ and $x < b$. Let us first observe that, if $x \geq t$, then the cost of $\mathsf{ALG}_t$ is $t + \mathsf{ALG}(x - t, b, Q_t)$, because a unit cost was paid during the first $t$ days, and then $\mathsf{ALG}$ is called provided with a prediction $Q_t$. The number of remaining snow days at that time is $x - t$. We remind that, since we are in a minimization problem, $C(\mathsf{ALG}, p_t)$ is at least 1. Consider the case where $x \leq b$, thus $\mathsf{OPT}(x, b) = x$. If $x \leq t$ then $\mathsf{ALG}_t(x, b) = x = \mathsf{OPT}(x, b)$, and if $x > t$ then

$$
\begin{aligned}
\mathbf{E}[\mathsf{ALG}_t(x, b)] &= t + \mathbf{E}[\mathsf{ALG}(x - t, b, Q_t)] \\
&\leq t + C(\mathsf{ALG}, p_t)\mathsf{OPT}(x - t, b) \\
&= t + C(\mathsf{ALG}, p_t)(x - t) \\
&= C(\mathsf{ALG}, p_t)x - (C(\mathsf{ALG}, p_t) - 1)t \\
&\leq C(\mathsf{ALG}, p_t)x - (C(\mathsf{ALG}, p_t) - 1)\frac{t}{b}x \\
&= \left( \frac{t}{b} + \left( 1 - \frac{t}{b} \right) C(\mathsf{ALG}, p_t) \right) \mathsf{OPT}(x, b) \\
&\leq \left( \frac{t}{b} + C(\mathsf{ALG}, p_t) \right) \mathsf{OPT}(x, b).
\end{aligned}
$$

Consider now that $x > b$, thus $\mathsf{OPT}(x, b) = b$. If $x \leq t$ then

$$
\mathsf{ALG}_t(x, b) = x \leq t \leq \left( \frac{t}{b} + C(\mathsf{ALG}, p_t) \right) \mathsf{OPT}(x, b).
$$

On the other hand, if $x > t$ then

$$
\begin{aligned}
\mathbf{E}[\mathsf{ALG}_t(x, b)] &\leq t + C(\mathsf{ALG}, p_t)\mathsf{OPT}(x - t, b) \\
&\leq t + C(\mathsf{ALG}, p_t)b \\
&= \left( \frac{t}{b} + C(\mathsf{ALG}, p_t) \right) \mathsf{OPT}(x, b).
\end{aligned}
$$

In all the cases we have $\mathbf{E}[\mathsf{ALG}_t(x, b)] \leq (t/b + C(\mathsf{ALG}, p_t))\mathsf{OPT}(x, b)$, which gives the result. $\square$

### A.2   Proof of Lemma 2.2

*Proof.* Let us denote $\mathsf{ALG}(x, b, \mathcal{Q})$ the output of Algorithm 1 given an instance $x, b$ and a prediction $\mathcal{Q}$, and $\mathsf{OPT}(x, b) = \min\{x, b\}$ the cost of the optimal offline algorithm. Purohit et al. [47] prove that, if the algorithm is given instead a prediction $y$ of $x$, then sets $\mathcal{Q} = \mathbb{1}(y \geq b)$ and behaves as we described, then its competitive ratio is upper bounded by $\min \left\{ 1 + \frac{1}{\lambda}, (1 + \lambda) + \frac{|x - y|}{(1 - \lambda)\mathsf{OPT}(x, b)} \right\}$. But in fact, it suffices to have $x$ and $y$ both larger or both smaller than $b$ for the algorithm to behave exactly as if $x = y$, therefore we retrieve the bound $1 + \lambda$ if the prediction $\mathcal{Q}$ is accurate. This happens with probability $p$. Otherwise, when the prediction is incorrect, we always have that

$\mathsf{ALG}(x, b, \mathcal{Q}) \le (1 + 1/\lambda)\mathsf{OPT}$. Therefore

$$\frac{\mathbf{E}[\mathsf{ALG}(x, b, \mathcal{Q})]}{\mathsf{OPT}(x, b)} \le p(1 + \lambda) + (1 - p)\left(1 + \frac{1}{\lambda}\right)$$

$$= 1 + p\lambda + \frac{1 - p}{\lambda},$$

the latter expression is minimized for $\lambda = \sqrt{\frac{1-p}{p}}$, which gives

$$\frac{\mathbf{E}[\mathsf{ALG}(x, b, \mathcal{Q})]}{\mathsf{OPT}(x, b)} \le 1 + 2\sqrt{p(1 - p)}.$$

$\square$

### A.3 Proof of Lemma 2.3

*Proof.* Let $\lambda \in (1/b, 1)$. It is shown in [47] that Algorithm 2 is $\left(\frac{\lambda}{1-e^{-\lambda}}\right)$-consistent and $\left(\frac{1}{1-e^{-(\lambda-1/b)}}\right)$-robust. Even though the upper bound shown in [47] suggests that the consistency bound is achieved only when given an exactly accurate prediction $y$ of the number of snow days, i.e. $y = x$, it suffices to have $\mathcal{Q} = \mathbb{1}(x \ge b)$ for the algorithm to behave just the same. Therefore, with probability $p$ we have that the cost of the algorithm is at most $\left(\frac{\lambda}{1-e^{-\lambda}}\right)\mathsf{OPT}(x, b)$, and with probability $1 - p$ it is at most $\left(\frac{1}{1-e^{-(\lambda-1/b)}}\right)\mathsf{OPT}(x, b)$. With a simple function analysis, we have that $\frac{1}{1-e^{-u}} \le \frac{1}{e-1} + \frac{1}{u}$ for any $u \in (0, 1)$. Therefore

$$\mathbf{E}[\mathsf{ALG}(x, b, \mathcal{Q})] \le \frac{p\lambda}{1 - e^{-\lambda}} + \frac{1 - p}{1 - e^{-(\lambda-1/b)}}$$

$$\le p\lambda\left(\frac{1}{e - 1} + \frac{1}{\lambda}\right) + (1 - p)\left(\frac{1}{e - 1} + \frac{1}{\lambda - 1/b}\right)$$

$$= \frac{p\lambda}{e - 1} + \frac{1 - p}{\lambda - 1/b} + \frac{1}{e - 1} + \left(1 - \frac{1}{e - 1}\right)p$$

$$= \frac{p(\lambda - 1/b)}{e - 1} + \frac{1 - p}{(\lambda - 1/b)} + \frac{1}{e - 1} + \left(1 - \frac{1 - 1/b}{e - 1}\right)p,$$

The previous expression is minimized when $\lambda$ satisfies $\frac{p(\lambda-1/b)}{e-1} = \frac{1-p}{(\lambda-1/b)}$, i.e. for $\lambda^\star = \frac{1}{b} + \sqrt{(e-1)\frac{1-p}{p}}$. However, we are restrained to choose $\lambda \in (1/b, 1)$, thus the optimal choice of $\lambda$ is $\min\{1, \lambda^\star\}$. We have that

$$\lambda^\star \le 1 \iff \frac{p(\lambda - 1/b)}{e - 1} = \frac{1 - p}{(\lambda - 1/b)} \le 1$$

$$\iff (e - 1)\frac{1 - p}{p} \le \left(1 - \frac{1}{b}\right)^2$$

$$\iff \frac{1}{p} \le 1 + \frac{(1 - 1/b)^2}{e - 1}$$

$$\iff p \ge \left(1 + \frac{(1 - 1/b)^2}{e - 1}\right)^{-1}.$$

Therefore, if $p \ge \left(1 + \frac{(1-1/b)^2}{e-1}\right)^{-1}$, then with $\lambda = \lambda^\star$ we have

$$\mathbf{E}[\mathsf{ALG}(x, b, \mathcal{Q})] \le \sqrt{\frac{p(1 - p)}{e - 1}} + \sqrt{\frac{p(1 - p)}{e - 1}} + + \frac{1}{e - 1} + \left(1 - \frac{1 - 1/b}{e - 1}\right)p$$

$$= \frac{1}{e - 1} + 2\sqrt{\frac{p(1 - p)}{e - 1}} + \left(1 - \frac{1 - 1/b}{e - 1}\right)p,$$

and if $p < \left(1 + \frac{(1-1/b)^2}{e-1}\right)^{-1}$, then with $\lambda = 1$ we obtain

$$
\begin{aligned}
\mathbf{E}[\mathsf{ALG}(x, b, \mathcal{Q})] &\leq \frac{p}{e-1} + \frac{1-p}{1-1/b} + \frac{1}{e-1} + \left(1 - \frac{1}{e-1}\right)p \\
&= \frac{1}{e-1} + \frac{1}{1-1/b} + \left(1 - \frac{1}{1-1/b}\right)p \\
&= \frac{1}{e-1} + \frac{b}{b-1} - \frac{1}{b-1}p \\
&= \frac{1}{e-1} + \frac{b-p}{b-1}.
\end{aligned}
$$

$\square$

## B  Secretary problem with $B$ predictions

### B.1  Proof of Theorem 3.1

*Proof.* Since the secretary problem is purely ordinal, and since the applicants arrive in a uniformly random order, the success probability of ADATHRESH does not depend on the values $x_1, \ldots, x_N$, but only on $B$ and $N$. Therefore, we denote $q_{N,B}$ the success probability of ADATHRESH with a budget $B$ and an input sequence of size $N$, $t^\star$ the position of the best applicant, $x_{s:t} = (x_s, \ldots, x_t)$ for any $s \leq t$, $S(x_{1:N}, B)$ the event that ADATHRESH with a budget $B$ succeeds on the input $x_{1:N}$, and $R_B = r_B N$. We define the random variable

$$
\tau = \min\left\{t \geq \lceil r_B N \rceil + 1 : x_t > \max\{x_1, \ldots, x_{t-1}\}\right\}.
$$

We will prove the claim of the theorem by induction on $B$.

For $B = 0$, it is established that for $r_0 = 1/\mathrm{e}$, we have a success probability of at least $q_0 = 1/\mathrm{e}$. Let $B \geq 1$, and assume that the result is true for $B - 1$, we have

$$
q_{N,B} = \mathbf{P}(S(x_{1:N}, B)) = \sum_{t=R_B}^{N} \mathbf{P}(S(x_{1:N}, B) \text{ and } \tau = t). \tag{3}
$$

For any $t$, since the oracle $\mathcal{Q}$ is error-free, then under $\tau = t$, we have that $\mathcal{Q}(x_t) = 0$ is equivalent to $t^\star = t$, and $\mathcal{Q}(x_t) = 0$ is equivalent to $t^\star > t$, thus we have

$$
\mathbf{P}(S(x_{1:N}, B) \text{ and } \tau = t \text{ and } \mathcal{Q}(x_t) = 0) = \mathbf{P}(\tau = t \text{ and } t^\star = t) = \frac{1}{N}\mathbf{P}(\tau = t \mid t^\star = t) \tag{4}
$$

$$
= \frac{1}{N}\mathbf{P}(\max x_{1:R_B} > \max x_{R_B+1:t-1}) \tag{5}
$$

$$
= \frac{1}{N} \times \frac{R_B}{t-1} = \frac{R_B/N}{t-1}, \tag{6}
$$

where 4 is true because the event $S(x_{1:N}, B)$ is implied by $\tau = t^\star$, the second equality because $t^\star$ is a uniform random variable in $\{1, \ldots, N\}$, and 5 because, conditionally to $t^\star = t$, the event $\tau = t$ is equivalent to not selecting any candidate in $x_{R_B+1:t-1}$, i.e all those candidates have smaller values than $\max x_{1:R_B}$, which is equivalent to saying that the maximum element in $x_{1:t-1}$ is among the first $R_B$ ones, and this is independent of the event $t = t^\star$. On the other hand, we have that

$$
\begin{aligned}
\mathbf{P}(S(x_{1:N}, B) \text{ and } \tau = t \text{ and } \mathcal{Q}(x_t) = 1) &= \mathbf{P}(S(x_{1:N}, B) \mid \tau = t, \mathcal{Q}(x_t) = 1) \\
&\quad \times \mathbf{P}(\tau = t \text{ and } t^\star > t).
\end{aligned}
$$

Conditionally to $\{\tau = t, \mathcal{Q}(x_t) = 1\}$, $x_t$ is the best candidate up to step $t$, and ADATHRESH used a prediction to observe that $t^\star > t$, The algorithm is therefore restarted with input $x_{t+1,N}$ and budget $B$, and $S(x_{1:N}, B) = S(x_{t+1:N}, B-1)$, which is independent of the history up to step $t$, hence

$$
\begin{aligned}
\mathbf{P}(S(x_{1:N}, B) \mid \tau = t, \mathcal{Q}(x_t) = 1) &= \mathbf{P}(S(x_{t+1:N}, B-1)) \\
&= q_{N-t, B-1} \geq q_{B-1}
\end{aligned}
$$

by the induction hypothesis. Secondly, we have that the two following events are equivalent

$$\{\tau = t \text{ and } t^\star > t\} = \{\max x_{t+1:N} > x_t > \max x_{1:R_B} > \max x_{R_B+1:t-1}\},$$

and since the applicants are shuffled uniformly at random, the relative ranks inside any subset $Y$ of $\{x_1, \ldots, x_N\}$ are independent of how $\max Y$ compares to applicants outside $Y$, therefore

$$\begin{aligned}
\mathbf{P}(\tau = t \text{ and } t^\star > t) &= \mathbf{P}(\max x_{t+1:N} > x_t > \max x_{1:R_B} > \max x_{R_B+1:t-1}) \\
&= \mathbf{P}(\max x_{1:N} \in x_{t+1:N})\mathbf{P}(\max x_{1:t} = x_t)\mathbf{P}(\max x_{1:t-1} \in x_{1:R_B}) \\
&= \frac{N-t}{N} \times \frac{1}{t} \times \frac{R_B}{t-1},
\end{aligned}$$

which gives

$$\mathbf{P}(S(x_{1:N}, B) \text{ and } \tau = t \text{ and } \mathcal{Q}(x_t) = 1) \geq \left(1 - \frac{t}{N}\right)\frac{R_B}{t(t-1)}q_{B-1}. \tag{7}$$

Using 4 and 7 we deduce that

$$\begin{aligned}
\mathbf{P}(S(x_{1:N}, B) \text{ and } \tau = t) &\geq \frac{R_B/N}{t-1} + \left(1 - \frac{t}{N}\right)\frac{R_B}{t(t-1)}q_{B-1} \\
&= (1 - q_{B-1})\frac{R_B/N}{t-1} + \frac{q_{B-1}R_B}{t(t-1)}
\end{aligned}$$

and substituting into 3 gives that

$$\begin{aligned}
q_{N,B} &\geq (1 - q_{B-1})\frac{R_B}{N}\sum_{t=R_B}^{N}\frac{1}{t-1} + q_{B-1}R_B\sum_{t=R_B}^{N}\frac{1}{t(t-1)} \\
&\geq (1 - q_{B-1})\frac{R_B}{N}\int_{R_B-1}^{N}\frac{du}{u} + q_{B-1}R_B\sum_{t=R_B}^{N}\left(\frac{1}{t-1} - \frac{1}{t}\right) \\
&= (1 - q_{B-1})\frac{R_B}{N}\log\left(\frac{N}{R_B-1}\right) + q_{B-1}\left(\frac{R_B}{R_B-1} - \frac{R_B}{N}\right),
\end{aligned}$$

since $R_B < r_B N + 1$, we have that $\log(N/(R_B - 1)) \geq \log(1/r_B)$, and

$$\frac{R_B}{R_B-1} - \frac{R_B}{N} \geq \left(1 + \frac{1}{R_B-1}\right) - \left(r_B + \frac{1}{N}\right) \geq 1 - r_B.$$

It follows that

$$q_{N,B} \geq (1 - q_{B-1})r_B\log(1/r_B) + q_{B-1}(1 - r_B) \tag{8}$$

The left term of this inequality is maximal for $r_B = \exp\left(-\frac{1}{1-q_{B-1}}\right)$, and we obtain that

$$\begin{aligned}
q_{N,B} &\geq (1 - q_{B-1})r_B \times \frac{1}{1-q_{B-1}} + q_{B-1}(1 - r_B) \\
&= r_B + q_{B-1}(1 - r_B) \\
&= q_{B-1} + (1 - q_{B-1})r_B = q_B,
\end{aligned}$$

which concludes the proof. $\qquad\square$

## B.2 Proof of Theorem 3.2

**The algorithm** Consider a simple adaptation of ADATHRESH, where the predictions are not always blindly trusted. If the conditions $t \geq r_B N$ and $x_t > \max\{x_1, \ldots, x_{t-1}\}$ are met, then with probability $1 - \lambda$ the algorithm chooses to query a prediction and trust it, and with probability $\lambda$ it chooses to select the candidate $x_t$ without querying a prediction.

We will prove Theorem 3.2 using two lemmas. In the first one, we show the lower bound on the success probability of the algorithm we presented, and in the second one, we compute the limit of this lower bound when $B \to \infty$. We define the function $f : [0, 1] \to (0, \infty)$ by $f(1) = 1$ and for $x \in (0, 1)$

$$f(x) = x + (1 - x)\exp\left(-\frac{1}{1-x}\right).$$

**Lemma B.1.** *The adapted* ADATHRESH *with budget $B$ has a success probability at least $q_B(p)$, where $q_0(p) = 1/e$ and for any $B \geq 1$*

$$q_B(p) = \max\{1/e, pf(q_{B-1}(p))\}.$$

*Moreover, if $p \leq (ef(1/e))^{-1}$ then for any $B$ we have $q_B(p) = 1/e$, and otherwise $q_B(p) = pf(q_{B-1}(p))$ for any $B \geq 1$.*

*Proof.* The success probability when $B = 0$ is lower bounded by $1/e$. Let $B \geq 1$ and assume that the success probability of this algorithm given a budget $B - 1$ is lower bounded by some constant $q_{B-1}(p)$ that is independent of $N$, if the algorithm chooses to trust the prediction, then if it inaccurate the algorithm fails with probability 1, and if it is accurate, which happens with a probability $p$, then similarly to (8) in the proof of Theorem 3.1, the success probability is lower bounded by

$$(1 - q_{B-1}(p))r_B \log(1/r_B) + q_{B-1}(p)(1 - r_B).$$

On the other hand, if the algorithm chooses not to trust the prediction, then its success probability is lower bounded by $1/e$. Therefore, if we denote $q_{N,B}(p, \lambda)$ the success probability of the algorithm on an instance of size $N$, given a budget $B$ of predictions that are each accurate with an independent probability $p$, then we have

$$q_{N,B}(p, \lambda) \geq \frac{\lambda}{e} + p(1 - \lambda)\left((1 - q_{B-1}(p))r_B \log(1/r_B) + q_{B-1}(p)(1 - r_B)\right)$$

The left term is maximized for $r_B = \exp\left(-\frac{1}{1-q_{B-1}(p)}\right)$, and it gives

$$q_{N,B}(p, \lambda) \geq \frac{\lambda}{e} + (1 - \lambda)p\left(q_{B-1}(p) + (1 - q_{B-1}(p))\exp\left(-\frac{1}{1-q_{B-1}(p)}\right)\right),$$

this is a linear function of $\lambda$, and it is maximized for

$$\lambda = \mathbb{1}\left(1/e > pq_{B-1}(p) + p(1 - q_{B-1}(p))\exp\left(-\frac{1}{1-q_{B-1}(p)}\right)\right).$$

With this choice of $\lambda$, we have that the success probability $q_{N,B}(p, \lambda)$ is lower bounded by the constant $q_B(p)$ defined by

$$q_B(p) = \max\left\{\frac{1}{e}, pq_{B-1}(p) + p(1 - q_{B-1}(p))\exp\left(-\frac{1}{1-q_{B-1}(p)}\right)\right\}$$

$$= \max\left\{\frac{1}{e}, pf(q_{B-1}(p))\right\}.$$

Before showing the second claim of the lemma, let us first prove that $f$ is increasing. We do this simply by computing its derivative

$$f'(x) = 1 - \left(\frac{1}{1 - x} + 1\right)\exp\left(-\frac{1}{1 - x}\right)$$

which is positive because $\exp(u) > u + 1$ for any $u > 1$, in particular for $u = 1/(1 - x)$.

Now, if $pf(1/e) \leq 1/e$, i.e. $p \leq (ef(1/e))^{-1} = \left(1 + (e - 1)\exp(-\frac{e}{e-1})\right)^{-1}$, since $f$ is increasing, we have by induction for any $B \geq 0$ that $q_B(p) = 1/e$. In fact, it is true for $B = 0$, and if $q_{B-1}(p) = 1/e$ then $pf(q_{B-1}(p)) = pf(1/e) \leq 1/e$, thus $q_B(p) = 1/e$.

On the other hand, if $pf(1/e) > 1/e$, then for any $B \geq 1$ we have that $pf(q_{B-1}(p)) > 1/e$ and thus $q_B(p) = pf(q_{B-1}(p))$. In fact, the property is true for $B = 1$, and if it is true for some $B \geq 1$ then $q_B(p) = pf(q_{B-1}(p)) > 1/e$, thus $pf(q_B(p)) \geq pf(1/e) > 1/e$, the property remains true for $B + 1$, which concludes the induction. $\square$

**Lemma B.2.** *If $p > (ef(1/e))^{-1}$, then the sequence $q_B(p)$ is increasing and*

$$\lim_{B \to \infty} q_B(p) = 1 - \frac{1}{1 + W\left(\frac{p}{e(1-p)}\right)},$$

*where $W$ is the inverse of $u \to ue^u$ on $[0, \infty)$, called the W-Lambert function .*

*Proof.* Assume that $p > (ef(1/e))^{-1}$, let us prove by induction over $B \geq 1$ that $q_B(p) > q_{B-1}(p)$. For $B = 1$ we have that $q_1(p) = pf(1/e) > 1/e = q_0(p)$ by the assumption on $p$. Now, let $B \geq 1$ and assume that $q_B(p) > q_{B-1}(p)$. Since $f$ is an increasing function on $[0, 1]$, we have that

$$q_{B+1}(p) = pf(q_B(p)) > pf(q_{B-1}(p)) = q_B(p).$$

The sequence $(q_B(p))_{B \geq 0}$ is therefore increasing and it is upper bounded by 1 since it is a lower bound on a probability. We deduce that it converges to some limit $\ell$ that depends on $p$. Since $f$ is continuous we have that $\ell = pf(\ell)$, solving this equation gives

$$
\begin{aligned}
\ell = pf(\ell) &\iff \ell = p\ell + p(1 - \ell) \exp\left(-\frac{1}{1 - \ell}\right) \\
&\iff (1 - p)\frac{\ell}{1 - \ell} = p \exp\left(-\frac{1}{1 - \ell}\right) \\
&\iff (1 - p)\left(\frac{1}{1 - \ell} - 1\right) = \frac{p}{e} \exp\left(1 - \frac{1}{1 - \ell}\right) \\
&\iff \left(\frac{1}{1 - \ell} - 1\right) \exp\left(\frac{1}{1 - \ell} - 1\right) = \frac{p}{e(1 - p)} \\
&\iff \frac{1}{1 - \ell} - 1 = W\left(\frac{p}{e(1 - p)}\right) \\
&\iff \ell = 1 - \frac{1}{1 + W\left(\frac{p}{e(1-p)}\right)} \ .
\end{aligned}
$$

Thereofore $\lim_{B \to \infty} q_B(p) = 1 - \frac{1}{1+W\left(\frac{p}{e(1-p)}\right)}$. $\qquad\square$

Observe that the algorithm we presented can also be adapted to the case where the accuracy $p$ of the oracle is not constant and might depend on the time or even on the input sequence. In fact, as long as the accuracy of the oracle is known to the algorithm, it can trust the oracle if it is correct with a probability larger than $(ef(1/e)^{-1}$, and ignore if otherwise. This gives a success probability always lower bounded by $1/e$, with better guarantees depending on the accuracy of the oracle at times when it is queried.

### B.3 ADATHRESH with memory

We present now an improved version of ADATHRESH that keeps in memory the best observed value so far, and that rejects all applicants below this value.

---

**Algorithm 5:** ADATHRESH with memory

---

**Input:** Budget $B$, sequence of applicants $(x_1, \ldots, x_N)$, largest previously observed value $M$
1 Reject the first $\lceil r_B N \rceil$ applicants;
2 **for** $t = \lceil r_B N \rceil + 1, \ldots, N$ **do**
3    **if** $x_t > M$ *and* $x_t > \max\{x_1, \ldots, x_{t-1}\}$ **then**
4       **if** $B > 0$ **then**
5          query a prediction $\mathcal{Q}(x_t)$;
6          **if** $\mathcal{Q}(x_t) = 0$ **then** Return: $t$ ;
7          **else** Return ADATHRESH$(B - 1, (x_{t+1}, \ldots, x_N), x_t)$;
8       **else** Return: $t$;

---

Initially, the algorithm is launched with $M = -\infty$. While we do not give theoretical guarantees on the success probability of Algorithm 5, we show experimentally how it compares to Algorithm 3 in Section 5.

## C  Preemptive $B$-clairvoyant job scheduling

### C.1  Proof of Theorem 4.1

*Proof.* Let ALG be any non-clairvoyant algorithm that can query up to $B$ job sizes for instances of $N$ jobs. It is shown in Theorem 2.8 in [46] that by choosing an input instance $\mathcal{E}_n$ of $n$ job sizes sampled independently from the exponential distribution, any algorithm A verifies

$$\mathbf{E}[\mathsf{A}(\mathcal{E}_n)] \geq (2 - 4/n)\mathbf{E}[\mathsf{OPT}(\mathcal{E}_n)],$$

with $\mathbf{E}[\mathsf{OPT}(\mathcal{E}_n)] = n + n(n + 1)/4$. Let us consider in particular the algorithm $\mathsf{A}_N$ which, when given an instance $\mathcal{J}$ of $n \leq N$ jobs, samples $N - n$ positive numbers $z_1, \ldots, z_{N-n}$ from the exponential distribution, then runs ALG with the input $\mathcal{J} \cup \{z_i\}_{i=1}^{N-n}$. Let us denote $\mathsf{A}_N(\mathcal{J}|\{z_i\}_{i=1}^{N-n})$ the output of $\mathsf{A}_N$ conditionally to $\{z_i\}_{i=1}^{N-n}$. Now, let $\mathcal{E}$ be an instance of $N$ job sizes chosen from the exponential distribution to be given as input for ALG. During its execution, ALG will query a number $b \leq B$ of sizes for jobs in some subset $I$ of $\{1, \ldots, N\}$. Denoting $J = \{1, \ldots, N\} \setminus I$, we have by Equation 1 that

$$\mathsf{ALG}(\mathcal{E}) \geq \sum_{i \in J} x_i + \sum_{i < j \in J} \left(D^{\mathsf{ALG}}(i, j) + D^{\mathsf{ALG}}(j, i)\right)$$
$$= \mathsf{A}_N\left(\{x_j\}_{j \in J}|\{x_i\}_{i \in I}\right).$$

In expectation, using that $|J| \geq N - B = N - o(N)$ with probability 1, we obtain

$$\mathbf{E}[\mathsf{ALG}(\mathcal{E})] \geq \mathbf{E}\left[\mathbf{E}[\mathsf{A}_N\left(\{x_j\}_{j \in J}|\{x_i\}_{i \in I}\right) \mid n = |J|]\right]$$
$$\geq \mathbf{E}\left[|J|^2/2 + 3|J|/2 - 5\right]$$
$$= \frac{N^2}{2} + O(N).$$

Using again that $\mathbf{E}[\mathsf{OPT}(\mathcal{E})] = N + N(N - 1)/4 = N^2/4 + O(N)$, we deduce that $\frac{\mathbf{E}[\mathsf{ALG}(\mathcal{E})]}{\mathbf{E}[\mathsf{OPT}(\mathcal{E})]} \geq 2 + o(1)$, which shows that the competitive ratio of ALG is at least 2. $\square$

### C.2  Intermediate Lemma and notation

**Notations**   In all the following, we fix an instance $\mathcal{J}$ of job sizes $x_1 \leq \ldots \leq x_N$, and we denote $\sigma$ a uniformly random permutation of $\{1, \ldots, N\}$, $I = \{\sigma(i)\}_{i=1}^{B}$ and $J = \{\sigma(j)\}_{j=B+1}^{N}$. This is equivalent to choosing a set $I$ of $B$ jobs uniformly at random without replacement. We also denote $I_t$ and $J_t$ respectively the sets of unfinished jobs in $I$ and $J$ at time $t$ of a run of PAR.

The next lemma gives a simple computational result that we will use in further proofs.

**Lemma C.1.** *For any $i \neq j$ we have* $\mathbf{E}[\min\{x_{\sigma(i)}, x_{\sigma(j)}\}] = \frac{2}{N(N-1)} \sum_{k=1}^{N} (N - k)x_k$.

*Proof.* Since $\sigma$ is a random permutation, we only need to demonstrate the result for $i, j = 1, 2$. Since we assume that $x_1 \leq \ldots, x_N$, then for any $k \leq N$ we have

$$\mathbf{P}(\min\{x_{\sigma(1)}, x_{\sigma(2)}\} = x_k) = \mathbf{P}(\min\{\sigma(1), \sigma(2)\} = k)$$
$$= 2\mathbf{P}(\sigma(1) = k \text{ and } \sigma(2) > k)$$
$$= 2\mathbf{P}(\sigma(1) = k)\mathbf{P}(\sigma(2) > k \mid \sigma(1) = k)$$
$$= \frac{2(N - k)}{N(N - 1)}.$$

It follows that

$$\mathbf{E}[\min\{x_{\sigma(1)}, x_{\sigma(2)}\}] = \sum_{k=1}^{N} \mathbf{P}(\min\{x_{\sigma(1)}, x_{\sigma(2)}\} = x_k)x_k$$
$$= \frac{2}{N(N - 1)} \sum_{k=1}^{N} (N - k)x_k.$$

$\square$

## C.3 Proof of Lemma 4.2

*Proof.* Let us fix some update rule for $\alpha$. In the execution of PAR, OPT is run on $I$ and RR on $J$, and both sets are disjoint, therefore we have $D^{\mathsf{PAR}}(i,j) = D^{\mathsf{OPT}}(i,j)$ for any $i \neq j \in I$, and $D^{\mathsf{PAR}}(i,j) = D^{\mathsf{RR}}(i,j)$ for any $i \neq j \in J$, we can write

$$\mathsf{PAR}(\mathcal{J}) = \sum_{i=1}^{N} x_i + D^{\mathsf{PAR}}(I,I) + D^{\mathsf{PAR}}(J,J) + D^{\mathsf{PAR}}(I,J) + D^{\mathsf{PAR}}(J,I)$$

$$= \sum_{i=1}^{N} x_i + D^{\mathsf{OPT}}(I,I) + D^{\mathsf{RR}}(J,J) + D^{\mathsf{PAR}}(I,J) + D^{\mathsf{PAR}}(J,I). \tag{9}$$

Using Equation 2 and Lemma C.1, we have that

$$\mathbf{E}[D^{\mathsf{OPT}}(I,I)] = \sum_{1 \leq i < j \leq B} \mathbf{E}[\min\{x_{\sigma(i)}, x_{\sigma(j)}\}]$$

$$= \sum_{1 \leq i < j \leq B} \frac{2}{N(N-1)} \sum_{k=1}^{N} (N-k)x_k$$

$$= \frac{B(B-1)}{N(N-1)} \sum_{k=1}^{N} (N-k)x_k,$$

and in the same way

$$\mathbf{E}[D^{\mathsf{RR}}(J,J)] = \sum_{B < i < j \leq N} 2\mathbf{E}[\min\{x_{\sigma(i)}, x_{\sigma(j)}\}]$$

$$= \frac{2(N-B)(N-B-1)}{N(N-1)} \sum_{k=1}^{N} (N-k)x_k$$

$$= \left(2 - \frac{2B(2N-B-1)}{N(N-1)}\right) \sum_{k=1}^{N} (N-k)x_k.$$

Summing these two equations gives

$$\mathbf{E}[D^{\mathsf{OPT}}(I,I)] + \mathbf{E}[D^{\mathsf{RR}}(J,J)] = \left(2 + \frac{B(B-1) - 2B(2N-B-1)}{N(N-1)}\right) \sum_{k=1}^{N} (N-k)x_k$$

$$= \left(2 - \frac{B}{N}\left(4 - 3\frac{B-1}{N-1}\right)\right) \sum_{k=1}^{N} (N-k)x_k,$$

and taking the expectation in Equation 9 concludes the proof. $\qquad\square$

## C.4 Proof of Theorem 4.3

We use here the same notations as in Section 4.3. Before demonstrating the theorem, let us justify why $(\alpha_t^{\mathsf{RR}})_t$ can be computed online, i.e for any $t > 0$, $\alpha_t^{\mathsf{RR}}$ can be computed knowing only the events that occurred up to time $t$.

For any subset $K$ of $\{1, \ldots, N\}$, we denote $\mathsf{RR}(K)$ the algorithm that runs RR on $\{x_i\}_{i \in K}$ and ignores the other jobs. Running RR on all the input instance is equivalent to running concurrently $\mathsf{RR}(I)$ and $\mathsf{RR}(J)$ with respective rates $(\alpha_t^{\mathsf{RR}})_t$ and $(1 - \alpha_t^{\mathsf{RR}})_t$. In fact, for any $t > 0$, the processing rate of any unfinished job in $I$ is $\frac{\alpha_t^{\mathsf{RR}}}{|I_t^{\mathsf{RR}}|} = \frac{1}{|I_t^{\mathsf{RR}}| + |J_t^{\mathsf{RR}}|}$, and we have the same for jobs in $J$.

Consider running RR on $\{x_1, \ldots, x_N\}$. We denote $t_i^{\mathsf{RR}}$ the completion time of job $1 \leq i \leq N$. For any $t > 0$, the amount of $\mathsf{RR}(I)$ executed until time $t$ is $\int_0^t \alpha_u^{\mathsf{RR}} du$. Therefore if we denote $T_i^{\mathsf{RR}}(I)$ the completion time of job $i \in I$ when running $\mathsf{RR}(I)$, then we have

$$T_i^{\mathsf{RR}}(I) = \int_0^{t_i^{\mathsf{RR}}(I)} \alpha_u^{\mathsf{RR}} du.$$

Knowing the sizes of jobs in $I$, $T_i^{\mathsf{RR}}(I)$ can be computed as

$$T_i^{\mathsf{RR}}(I) = x_i + \sum_{\substack{j \in I \\ j \neq i}} D^{\mathsf{RR}}(j,i) = x_i + \sum_{\substack{j \in I \\ x_j \leq x_i}} x_j + \sum_{\substack{j \in I \\ x_j > x_i}} x_i.$$

We deduce that $\mathsf{PAR}$ with SRR update rule can be implemented running concurrently $\mathsf{OPT}(I)$ and $\mathsf{RR}(J)$ with respective rates $\alpha^{\mathsf{RR}}$ and $1-\alpha^{\mathsf{RR}}$, where the processing rate updates as follows: initialize $c(I) = B$, $c(J) = N - B$ and $\alpha^{\mathsf{RR}} = \frac{c(I)}{c(I)+c(J)} = \frac{B}{N}$, and for any $t > 0$,

(i) if a job in $J$ terminates, then $c(J) \leftarrow c(J) - 1$,

(ii) if $T_i^{\mathsf{RR}}(I) = \int_0^t \alpha_u^{\mathsf{RR}} du$ for some $i \in I$, then $c(I) \leftarrow c(I) - 1$,

and in both cases update $\alpha^{\mathsf{RR}} = \frac{c(I)}{c(I)+c(J)}$.

*Proof of Theorem 4.3.* Using the processing rate $(\alpha_t^{\mathsf{RR}})_{t>0}$, the delays generated by $I$ on $J$ are exactly the same as if $\mathsf{RR}$ was running, and the delays by $J$ on $I$ are smaller than $\mathsf{RR}$'s, because $\mathsf{OPT}(I)$ runs with the same processing rate as $\mathsf{RR}(I)$ but all the jobs finish earlier, thus using Lemma C.1 we deduce that

$$\mathbf{E}[D^{\mathsf{PAR}}(I,J) + D^{\mathsf{PAR}}(J,I)] \leq 2\sum_{i=1}^{B}\sum_{j=B+1}^{N} \mathbf{E}[\min\{x_{\sigma(i)}, x_{\sigma(j)}\}]$$

$$= \frac{4B(N-B)}{N(N-1)} \sum_{k=1}^{N}(N-k)x_k.$$

Using Lemma 4.2 then Equation 2 we obtain

$$\mathbf{E}[\mathsf{PAR}(\mathcal{J})] \leq \sum_{i=1}^{N} x_i + \left(2 - \frac{B}{N}\left(4 - 3\frac{B-1}{N-1}\right) + \frac{4B(N-B)}{N(N-1)}\right) \sum_{k=1}^{N}(N-k)x_k$$

$$= \sum_{i=1}^{N} x_i + \left(2 - \frac{B(B-1)}{N(N-1)}\right) \sum_{k=1}^{N}(N-k)x_k$$

$$\leq \left(2 - \frac{B(B-1)}{N(N-1)}\right)\left(\sum_{i=1}^{N} x_i + \sum_{k=1}^{N}(N-k)x_k\right)$$

$$= \left(2 - \frac{B(B-1)}{N(N-1)}\right)\mathsf{OPT}(\mathcal{J}).$$

Which proves the theorem. $\qquad\square$