# OpenReview forum: "Advice Querying under Budget Constraint for Online Algorithms"
_NeurIPS.cc/2023/Conference — NeurIPS 2023 poster_

### Official Review · Reviewer_HbqC · 2023-07-04

**Soundness:** 2 fair
**Presentation:** 2 fair
**Contribution:** 2 fair
**Rating:** 5
**Confidence:** 4

**Summary:**

This paper considers learning augmented online algorithms where there is a bound on the number of predictions that can be queried.  Three problems are considered within this framework: Ski-rental, Secretary Problem, and Non-clairvoyant scheduling.

For the ski-rental problem, they consider a model where the algorithm can make noisy queries of the form "$x-t  > b$?", where $t$ is the current day.  The output of the query is a binary value which is correct independently with probability $p_t$, and the sequence $p_t$ is known up front to the algorithm.  Modifying ideas from Purohit et al. NeurIPS 2018, they give deterministic and randomized algorithms whose competitive ratio improve as the probability of correctness approaches 1 and are never worse than 2 and $e/(e-1)$, respectively.

For the secretary problem, they again consider predictions which are correct with probability $p$, but the algorithm can only query the predictions at most $B$ times.  Here the prediction is a binary prediction of whether or not the current item has the highest value in the sequence.  They give an algorithm which selects the best item with probability $1/e$ and improves as the probability $p$ increases.

For non-clairvoyant scheduling, the paper defines the problem of $B$-clairvoyant scheduling, in which the scheduler can query the exact sizes of $B \leq N$ jobs.  For this they show that $B = \Omega(N)$ queries are necessary to improve the competitive ratio below 2 which is met by the round robin algorithm in the non-clairvoyant setting.  To complement this they give an algorithm with competitive ratio $2-B(B-1)/(N(N-1))$, which gives an interpolation between the non-clairvoyant and fully-clairvoyant settings.

Finally, a short experiment section which tests the proposed algorithms on synthetic data is provided.

**Strengths:**

With the recent interest in learning-augmented algorithms/algorithms with predictions there has been recent interest in studying the extent to which such algorithms need to access to many predictions in order to achieve better than worst case results, so the considered problems are well motivated.

The results on the secretary problem and non-clairvoyant job scheduling are interesting.  In particular, the lower bound (Theorem 4.1) holds even in the case where the predictions match the ground truth.  The overall result presents a clean interpolation between the clairvoyant and completely non-clairvoyant settings.



**Weaknesses:**


- For the ski-rental result, the authors claim that they give an algorithm which chooses the best time to query the prediction based on the sequence $p_t$ (lines 79-80).  From what is presented in Section 2. it is not at all clear that they do this.  At best, there is an example for a specific sequence of $p_t$'s, but I don't see a general argument.

 - In terms of novelty, the techniques for ski-rental and non-clairvoyant scheduling can be viewed as extensions of the ideas presented in Purohit et al. NeurIPS 2018.

- There is a disconnect between the first two sections (ski-rental, secretary) and the third (scheduling).  For the first two the predictions are only correct with some probability while for the scheduling result the queries return exact job size values.  A more unified framework would make the paper significantly cleaner.

 - Aside from the scheduling result, the authors do not give evidence for the tightness of their results.

- The experiments mostly confirm what is predicted by the theory and don't seem to say much else.  Experiments on real data would alleviate this and also provide more evidence for the assumption that predictions are correct independently with some known probability.

**Questions:**

 - Concerning the lower bound for B-clairvoyant scheduling (Theorem 4.1), I think one of the references at the beginning of the proof is wrong.  For the lower bound of any possibly randomized algorithm against jobs independently sampled from an exponential distribution, the reference should be for Theorem 2.8 in [44] (not [45]).

- There is other recent work which considers learning-augmented online algorithms with limited querying.  See the following:
     - Im, S., Kumar, R., Petety, A., Purohit, M.. (2022). Parsimonious Learning-Augmented Caching. Proceedings of the 39th International Conference on Machine Learning, in Proceedings of Machine Learning Research 162:9588-9601 Available from https://proceedings.mlr.press/v162/im22a.html.

- For ski-rental and the secretary problem, this paper considers predictions which are correct independently with some probability $p$.  A similar model has also been considered recently for caching and online covering problems.  See the following:
    - Gupta, Anupam and Panigrahi, Debmalya and Subercaseaux, Bernardo and Sun, Kevin.   Augmenting Online Algorithms $\varepsilon$-Accurate Predictions.  Advances in Neural Information Processing Systems 2022.

- For the experiments on $B$-clairvoyant scheduling, it seems it would be natural to consider exponentially distributed job sizes, since those witness the lower bound for Theorem 4.1.

- For the ski-rental and secretary problem results, is it possible to give any general lower bounds for the competitive ratio of any algorithm?  For the ski-rental result this could help to alleviate the concerns the authors raise with using the simpler strategy over the optimal strategy which is difficult to analyze.

- Please use proper capitalization for the title and section titles.

**Limitations:**

This paper is mostly theoretical in nature, so any possible limitations are inherent in the assumptions and theorem statements which have been stated clearly (and thus addressed by the authors).

---

> ### Author Rebuttal · Authors · 2023-08-08
>
> We thank you for reviewing our submission and for your feedback. Below, we address the weaknesses and questions that you raised.
>
> ## Weaknesses
> - We explain how to choose the time of querying the prediction in the discussion following Lemma 2.1 (paragraph starting in Line 165). Given a learning-augmented algorithm with a competitive ratio $C(p)$ that depends on the accuracy $p$ of the predictor, Lemma 2.1 shows how to deduce an algorithm with a competitive ratio of $t/b + C(p_t)$ for any $t\geq 0$. The best time to query the prediction is, therefore, $t^\star$ minimizing the function $t \mapsto t/b + C(p_t)$. It depends on the functions $p\to C(p)$ and $t \to p_t$.
>
>
> - We respectfully disagree with this statement:
>     - Regarding the ski-rental problem, it is true that **we use as an example the algorithms introduced by Purohit et al.,** however, the techniques we used in the proofs are different, and the main idea we presented is how to estimate the competitive ratio of these algorithms when the accuracy of the predictor is known, how to adjust $\lambda$ in such case, and how to choose the time of querying the prediction. This is a general idea that can also be used with any other learning-augmented algorithm taking a binary prediction as input.
>     -  Secondly, regarding the scheduling problem, **the setting we investigate, results, and proofs are independent of those in the paper of Purohit et al.** The only resemblance is that we run the jobs in parallel with appropriate processing power each, and this is a core idea in all preemptive scheduling algorithms (for example RR algorithm). In the problem we proposed, the difficulty is not to deal with inaccurate predictions, but instead to find a way to make good use of a limited number of perfect hints, which is not trivial.
>
> - See the common answer.
>
> - See the common answer.
>
> - The experiments indeed confirm the theoretical results [as expected], but they also provide additional insights. For the secretary problem, we only showed a theoretical lower bound on the success probability of ADATHRESH, while Figure 5 shows that this lower bound is actually tight when $N$ is large. Also, the same figure shows that the heuristic ADATHRESH with memory has a better success probability. On the other hand, Figure 6 shows that with many natural and benchmark instances, the algorithm we proposed has a competitive ratio $2-B/N$, which is better than the theoretical bound $2-B(B-1)/(N(N-1))$.
>
> ## Questions
> - Yes, thank you for noticing it. We will correct the reference.
>
> - We agree, it is a closely related paper, as they consider the caching problem with a limited budget of predictions. We will include it in the related work.
>
> - Their model is indeed similar to the one studied here in spirit. We will cite them in the related work section.
>
> - We considered exponentially distributed job sizes in Figure 6 (orange curve, (ii) in the discussion on the figure).
>
> - See the common answer.
>
> - Thank you for the remark, we will address any errors related to capitalization in the title and section titles.

---

> > ### Comment · Reviewer_HbqC · 2023-08-16
> >
> > Thank you for your response.  I see now the way to choose a query time, although I think further clarity on this within the paper would be helpful to the reader.  I have raised my score slightly.

---

### Official Review · Reviewer_imY3 · 2023-07-05

**Soundness:** 4 excellent
**Presentation:** 4 excellent
**Contribution:** 3 good
**Rating:** 8
**Confidence:** 4

**Summary:**

The paper studies learning-augmented algorithms in a setting where predictions are a scarce resource and algorithms need to decide when to request them. The framework is instantiated over (i) ski-rental with a single prediction, (ii) the secretary problem with B predictions, (iii) preemptive non-clairvoyant scheduling with B perfectly accurate predictions. The article mainly presents upper bounds for these problems.

**Strengths:**

- The paper is really well-written
- The framework is well-motivated and likely to inspire further work
- The results are nice and non-trivial
- Proofs (as far as I checked) in the supplementary material seem to be correct
- Experimental figures help understanding the parameters in the results (e.g., roles of $b$ or $p$)

**Weaknesses:**

No major weaknesses, so I'll point some relatively minor ones.

1) It's unclear how novel the setting is and I think the paper could be more straightforward about this; the authors provide a section on related work, but while they mention the setting is relatively unexplored, there are examples of the setting that they provide as well as others they don't seem to cite like "Parsimonious Learning-Augmented Caching" by Im et al. The abstract of the paper, as well the introduction page, give me the feeling that the authors are presenting the setting itself as a contribution, whereas it seems that setting is not itself novel and the contribution is about the specific results for specific problems presented (which is not minor).

2) On Algorithm 2 I think it would be nicer to write "buy on the start of day $d \in ...$ with probability proportional to $(1-1/b)^{\lfloor \lambda b \rfloor - d}$.

3) The results (i.e., theorem statements) could be phrased in terms of consistency/robustness more explicitly.

4) As usual I printed the paper on black-and-white, and the figures are not great in this respect -- consider for example Figure 1, where the lines only differ by color. I suggest using either some pattern (dashed, dotted, etc, (Figure 2 does a bit of this, but it can still be improved)). Granted this is not an issue on a computer screen, but it's worth considering too for accessibility reasons (colorblind readers, etc.)

5) The scheduling problem setting is a bit weird as the predictions are assumed to be perfect, and thus said problem doesn't really fit the learning-augmented framework, only the budget-constrained aspect. It would be more solid if the paper considered predictions with a parametric quality.

6) It seems that the predictions considered are always in terms of their probability of being correct rather than on a notion of error (as in Lykouris and Vassilvitskii), which fits the framework of (https://openreview.net/forum?id=HFkxZ_V0sBQ), and seems appropriate to cite.



**Questions:**

For the ski-rental problem, did you consider what happens if one has B predictions that are independent? An interesting thing that might happen is that two general strategies appear to be in conflict: (1) Spend the B predictions at the beginning to have as good as possible an initial prediction, due to the independence assumption. (2) Wait until predictions unfold (i.e., are confirmed or refuted by reality) and when the season continues spend more predictions to sort of recursively reset the state imagining the season just started.



**Limitations:**

They're not really addressed, although I don't see an issue in the particular case of this paper as there doesn't seem to be much to address anyway.

---

> ### Author Rebuttal · Authors · 2023-08-08
>
> We thank you for reviewing our paper, and for your positive evaluation of our work. We are very happy that you appreciated the paper and enjoyed reading it. Below we address the weaknesses and questions you raised.
>
>
> ## Weaknesses
> **1.** Regarding the paper by Im et al., it is indeed a closely related paper to our work, as they consider the caching problem with a limited budget of predictions. We will update the related work section to include it, and we will try to better explain the contribution in the introduction.
>
> **2.** Yes this indeed is a nicer way to write the algorithm. Thank you for the suggestion.
>
> **3.** We will be more explicit about them in the results statements.
>
> **4.** We completely agree, we will update the figure and take this into account in the other figures also.
>
> **5.** See the common answer.
>
> **6.** The predictions in the problems we studied are indeed considered in terms of their probability of being true. We will cite the work of Gupta et al. in the related work. It is also worth mentioning that Lemmas 2.2 and 2.3 in our paper make a transition (in the particular case of binary predictions) between the framework of Vassilvitskii and that of Gupta et al.
>
>
> ## Questions
> Even more generally, consider having access to a group of $B$ experts, each of whom can only be consulted once during the algorithm's execution. Each expert $b \in \{1,\ldots,B\}$ has an independent probability $p^b_t$ of being correct, which varies with time.
> As you mentioned, several strategies can be used. The first approach is to select a specific time to query all the experts' predictions simultaneously, hence obtaining a more accurate recommendation. Alternatively, a second strategy involves using the predictions in a sparse manner. Lastly, there are intermediate strategies where a few predictions are used at the beginning (instead of just one), then a few others at some ulterior step, and so forth. Given the experts' different accuracy $p^b_t$ for $1 \leq b \leq B$, the order in which the algorithm queries their advice becomes crucial.
> This is an intriguing problem that we are exploring and plan to include in the journal version of the paper.

---

### Official Review · Reviewer_rY4H · 2023-07-06

**Soundness:** 3 good
**Presentation:** 3 good
**Contribution:** 2 fair
**Rating:** 4
**Confidence:** 4

**Summary:**

Authors consider ML-augmented algorithms with a limited number of predictions.
They consider three problems: ski rental, secretary, and non-clairvoyant
scheduling. Their result on non-clairvoyant scheduling assumes perfect
predictions, while for ski rental and secretary problems they assume
knowledge of the predictors success probability in advance.
This implicitly assumes that the input has a particular structure or comes from
a distribution where the predictors achieve the declared success probability.
In case of the secretary problem, they also discuss the case when an incorrect
value of p is provided to the algorithm. I have not noticed such a discussion
in the case of ski rental.
It is not clear whether the performance bounds achieved by their algorithms
are tight.


**Strengths:**

technically sound results

**Weaknesses:**

* absence of lower bounds for ski rental and secretary
* only perfect predictions for scheduling
* not clear how should we know the success probability of the predictor

**Questions:**

* there is a paper by Antoniadis et al. Learning-Augmented  Dynamic Power Management with Multiple States via New Ski Rental Bounds (NeurIPS 2021) which, as a special case, solves a sequence of ski rental instances, learning the precision of the predictor online. Can you relate somehow to their results?
* I appreciate Figure 2 showing your competitive ratio in a natural setting. However, it looks like your ratio is never better than e/(e-1). What precision do you need to achieve meaningful improvement?
* There is a paper by Im et al. Parsimonious Learning-Augmented Caching (ICML 2022) -- might be also worth mentioning in your related work.

**Limitations:**

theorems state their assumptions clearly

---

> ### Author Rebuttal · Authors · 2023-08-08
>
> We thank you for reviewing our submission and for your feedback. Below, we address the weaknesses and questions that you raised.
>
> ## Weaknesses
> See the common answer.
>
> ## Questions
> - The paper by Antoniadis et al. studies the problem of dynamic power management (DPM), which is a generalization of the ski-rental problem. They introduce a notion of $(\rho,\mu)$-competitiveness, where $\rho$ indicates the consistency of the algorithm and $\mu$ describes the dependence on the prediction error. Then, they design a near-optimal algorithm in the sense of this definition. The setting we study and our results are different from theirs, and we do not see how to relate or compare both papers. However, since the learning-augmented DPM problem involves multiple unknown variables, requiring a prediction for each, it would be very interesting to investigate it when the number of predictions is limited.
>
>
> - The blue curves represent the competitive ratio of the algorithm that queries a prediction at time $t$ and then runs the deterministic Algorithm 1. If the prediction is queried at $t=0$ then we find the optimal deterministic algorithm without prediction, which has a competitive ratio of $2$. The figure shows that a competitive ratio better than $2$ can be achieved by optimally choosing the time of querying the prediction.
> The orange curves represent the competitive ratio achieved by querying a prediction at time $t$ and then running the randomized Algorithm 2, and these are the ones that should be compared to $e/(e-1) \approx 1.58$. More precisely, in the setting without predictions, the competitive ratio of any algorithm is at least $\frac{1}{1-(1-1/b)^b}$, which is very close to $e/(e-1)$ for $b = 50$ and $b=100$.  **The dashed curve ($b=100$) goes indeed below** $e/(e-1)$, beating the worst-case without prediction, while **the continuous one ($b=50$) always stays above it**, and equals the worst-case competitive ratio without prediction only when the prediction is queried at $t=0$. We chose the values of $b$ and the function $t \to p_t$ on purpose to illustrate that, depending on the problem's parameters, it can be optimal to query the prediction at $t=0$ instead of querying it at an ulterior time.
>
>
> - Yes indeed, this paper is closely related to ours, as they study the caching problem with a limited budget of predictions. We include it in the related work.

---

> > ### Comment · Reviewer_rY4H · 2023-08-12
> >
> > thank you for your explanation. regarding Figure 2, you have not said what is actually your competitive ratio. The bottom of the plot seems to correspond to competitive ration 1.5, this is barely better than 1.58 and you did not respond to my question about the precision required to get something significantly better. But in any case, I found weaknesses mentioned in my review important, also I do not really like your model which implicitly assumes pretty strong stochastic properties about the input in order to allow receive predictions with probability corresponding to the described curve (at least you did not object to this statement in my review). Therefore I do not change my rating.

---

> > > ### Author Response · Authors · 2023-08-13
> > > **Weaknesses are addressed in the global response**
> > >
> > > Thank you for reading our rebuttal and for your response.
> > >
> > > ### Weaknesses
> > > Regarding the weaknesses, **please read the “global response” addressed to all the reviewers**. We referred to this global response in the rebuttal we wrote for your review “Weaknesses: see the common answer”. Here are the paragraphs objecting to each weakness in the global response
> > > - **absence of lower bounds for ski rental and secretary** See the paragraphs “Ski-rental: Tightness of the results” and “Secretary: Tightness of the results”.
> > > - **only perfect predictions for scheduling** See the paragraph “Why is the advice considered to be perfect for the scheduling problem?”.
> > > - **not clear how should we know the success probability of the predictor** See the paragraphs “Ski-rental: How to know the accuracy of the oracle?” and “Secretary: How to know the accuracy of the oracle?”.
> > >
> > > ### Questions
> > > Regarding Figure 2, the minimal competitive ratio is 1.51. With the same function $p_t$, if $b=500$ then the minimal competitive ratio becomes 1.36, which is significantly better than $e/(e-1)$. As precised in the rebuttal, it depends both on $b$ and $t \mapsto p_t$. The improvement depending on the precision of the predictor is shown in Figure 1.

---

### Official Review · Reviewer_gTsH · 2023-07-06

**Soundness:** 4 excellent
**Presentation:** 3 good
**Contribution:** 4 excellent
**Rating:** 7
**Confidence:** 4

**Summary:**

In this paper the authors re-consider classic online optimization problems in the area of learning-augmented algorithms, under the setting where the algorithms that have access to a limited number of predictions. Additionally, the setting that is considered assumes that as the algorithm progresses the predictions become more accurate. Three problems are studied under this model: 1) the ski-rental problem, 2) secretary problem, and 3) job scheduling problem. Each of these problems considers problem-specific advice, and the authors show that given correct advice they can significantly outperform the worst-case bounds from the non-learning-augmented model.

The main motivation for this work is that the advice can be very costly, and one cannot always assume access to unlimited calls to the learned advice. Hence, it becomes relevant to ask what are the best-possible bounds that one can obtain under this limited budget setting.

While the paper is mainly interesting for its theoretical contributions, the authors additionally ran simulations showing that the algorithms behave according to the theoretical results. Also for the secretary problem the authors show a heuristic variation of the suggested algorithm that performs better than the theoretical lower bound.

**Strengths:**

+ Considers an interesting setting where the algorithm has a limited access to an oracle, and in one where the probability of correct advice increases over time. Although, it is not clear how to motivate the very restricted access (why would it be restricted if it comes from a model).

+ A nice set of results for each of the three problems considered.

**Weaknesses:**

- Some advice and assumptions seem a bit unrealistic. Specifically, the advice on the size of the jobs in the job scheduling problem assumes that the advice is always correct but it is not clearly justified. Additionally, to tune the algorithms for the ski-rental problem, one needs to have knowledge of the probability of a correct prediction over time.

**Questions:**


- For the Secretary problem, it is not clear what it means that "assuming that the predictions are error-free, the problem is equivalent to the multiple-choice secretary problem". In the setting of the paper, the algorithm has to additionally choose whether to select the secretary or keep searching.

- The algorithm for the ski-rental problem assumes the knowledge of the success probability. Is this something realistic?

- L226: "The previous theorem shows that, with well-chosen thresholds for querying the predictions, strong theoretical guarantees on ADATHRESH can be obtained". This is not obvious and should be explained.

- Explain how the memorization variant of the algorithm for the secretary problem works.

- No discussion on the experiment for the ski-rental problem.

- Don't know how to read figure 4. Please explain.

- Report standard deviation for the plot with the scheduling problem.

---

> ### Author Rebuttal · Authors · 2023-08-08
>
> We thank you for the time and effort spent on our submission and for the positive evaluation of our work. We address below the weaknesses and questions that you raised.
>
> ## Weaknesses
> See the common answer.
>
> ## Questions
>
> - In the secretary problem with perfect advice, as you explained in your comment, the algorithm has to additionally choose whether to select the secretary or keep searching. However, **after the perfect advice is given, the decision to make is trivial**: stop if the secretary is the best overall, and keep searching otherwise.
> We can formally prove this equivalence as follows:
>      -  If $A$ is an algorithm for the multi-choice secretary problem with $B+1$ choices, then for a given instance $x_1,\ldots, x_N$, we can define an algorithm $\bar{A}$ for the secretary problem with $B$ perfect hints such that, at any step $t$ where $A$ selects a secretary, if $\bar{A}$ has a null budget, then it accepts the secretary, and otherwise it queries a hint telling if $x_t$ is the best overall and if it is the case then $\bar{A}$ accepts it and stops. Both $A$ and $\bar{A}$ succeed if one of the secretaries selected by $A$ is the best overall.
>      - On the other hand, if $\bar{A}$ is an algorithm for the secretary problem with $B$ perfect hints. Let $A$ the algorithm for the $B+1$-choice secretary problem defined as follows: if $\bar{A}$ queries advice for some candidate $x_t$, then select them, but since $A$ cannot have immediate access to the feedback, it assumes that $\bar{A}$ was advised to reject the candidate and continues. Finally, when $\bar{A}$ exhausts all its budget, $B$ candidates will have been selected by $A$, and the last one selected is also the last candidate that $\bar{A}$ selects without asking for advice since it has no more budget. With this, $A$ succeeds on a given instance if and only if $\bar{A}$ succeeds on the same instance.
>
> - See the common answer
>
> - The sequence $q_B$ is increasing and upper bounded by $1$, thus it converges to some limit $\ell$. Using the recurrence defining $q_B$ we find that $\ell = 1$. This can be seen as an extension of the limit in Theorem 3.2 when $p$ goes to $1$. However, we agree that it should be mentioned right after Theorem 3.1.
>
> - ADATHRESH is defined recursively. If it is called at step $t$, then it runs on the instance $(x_{t+1},\ldots,x_N)$, losing the information of $m = \max(x_1,\ldots,x_t)$. The memorization variant keeps this information in memory when restarted, and it skips all the candidates that have values below $m$. For candidates having a value larger than $m$, it behaves as ADATHRESH without memorization.
>
> - We present and discuss experiments for the ski rental problem in Appendix D. We will add them to the camera-ready version as an additional content page is allowed.
>
> - Figure 4 compares a run of RR (round-robin) and PAR with SRR update rule on the same instance. Let us ignore the yellow color for now. We consider an instance of 5 jobs indexed as 1,2,3,4,5. **Time is represented on the X-axis, and the Y-axis indicates the processing power allocated to each job**. In the left figure, we show a run of RR on this instance: initially, all 5 jobs run with a processing power 1/5 each. the first vertical line indicates the time when job 1 terminates. After that, the remaining jobs (2,3,4,5) run each with a processing power of 1/4 until job 2 is terminated, and so on. Now the yellow color is used to show the set $I = \{1,2,3\}$, which are the jobs with known sizes in this example.
> The yellow area in the right figure is identical to that of the left figure, and at any time, its height indicates the total processing power allocated to $I$ by SRR.
> Thank you for your remark on this Figure. We already updated it by indicating that the X-axis represents time and that the Y-axis indicates the processing power.
> - We will report standard deviations in Figure 6 for the camera-ready version.

---

> > ### Comment · Reviewer_gTsH · 2023-08-16
> >
> > Thank you for your explanations. I'm mostly satisfied with the explanations the authors provided (maybe a bit hard to buy the setting with the assumed perfect predictions for the scheduling problem), and will keep my recommendation for acceptance.

---

### Author Rebuttal · Authors · 2023-08-08


We would like to thank the reviewers for their feedback and thoughtful comments toward improving our paper. We address below some general comments and weaknesses that were raised by the reviewers.

### Ski-rental: Tightness of the results
We do not provide tightness results for the ski-rental problem, but we plan on working on them for the journal version of the paper. The study of tightness and trade-offs between consistency and robustness in learning-augmented algorithms can be a difficult task, for instance it is the main topic of many papers such as "Optimal Robustness-Consistency Trade-offs for Learning-Augmented Online Algorithms" (NeurIPS 2020).


### Secretary: Tightness of the results
We did not prove an upper bound because the main focus of the paper, in the section on the secretary problem, is to design an algorithm taking advantage of the predictions, even if inaccurate, to have a success probability better than $1/e$. Moreover, as we discuss in the paragraph starting in Line 252, if the predictions are accurate, then the problem we described is equivalent to the multi-choice secretary problem (Gilbert and Mosteller, 1966), and we are not aware of any upper bounds for the problem that we can try to extend in the case of inaccurate predictions.

### Ski-rental: How to know the accuracy of the oracle?
For the settings modeled as variants of the ski rental problem, it is possible in many cases to have a good estimate of the accuracy of the predictions. Take for example weather forecasts, the newsletter "forecast performance 2021" of ECMWF (European Centre for Medium-Range Weather Forecasts) shows that the quality of predictions, as a function of the lead time before the day concerned by the prediction, can be estimated very precisely.
On the other hand, the bounds we provide hold also if the accuracy of the predictions is at least $p_t$, which is a weaker condition.


### Secretary: How to know the accuracy of the oracle?
This assumption might be easier to understand with an "ordinal decision maker" and a "cardinal oracle". More precisely, consider a decision-maker that has an imperfect interviewing process, and thus can only compare the applicants with each other, i.e. observe their relative ranks, but lacks access to their true value. On the other hand, an expert who, due to extensive experience, knows how to better interview the applicants and extract their true value, and also knows the distribution of their values [like in a prophet setting].
The decision-maker can ask the expert to interview some candidates for a cost. After that, the expert does not disclose the precise value of the applicant - that is irrelevant to the decision maker-, but they provide a recommendation (accept or reject) and accompany it with a confidence level, which is the probability that this recommendation is accurate.


### Why is the advice considered to be perfect for the scheduling problem?
The clairvoyant and non-clairvoyant scheduling problems both have many real-life applications. It is therefore easy to imagine applications for the $B$-scheduling problem, which is an in-between setting. In opposite to the ski rental or the secretary problem, where the advice is a forecast/prediction, the advice provided in the scheduling problem can be perfect in many cases.
Moreover,  we demonstrate by studying this problem that 1) even using perfect advice can be non-trivial, 2) and also that the decision-maker cannot always take advantage of the ability to choose the time to query advice instead of having it as input, as the optimal time to query it, in this problem, is at the beginning.

---

### Decision · Program_Chairs · 2023-09-21

**Decision:**

Accept (poster)

**Comment:**

This paper was generally liked by the reviewers. The framework was viewed as motivated and could be of use in the future. The results were viewed as a complete story and expanded the current literature. The authors are encouraged to improve the writing and comparison to prior work as pointed out by the reviewers.